# Meta-analysis of sub-Saharan African studies provides insights into genetic architecture of lipid traits

Ananyo Choudhury [1,16✉], Jean-Tristan Brandenburg [1,16], Tinashe Chikowore[1,2], Dhriti Sengupta[1],
Palwende Romuald Boua [1,3], Nigel J. Crowther[4], Godfred Agongo[5,6], Gershim Asiki [7],
F. Xavier Gómez-Olivé [8], Isaac Kisiangani[7], Eric Maimela[9], Matshane Masemola-Maphutha[10],
Lisa K. Micklesfield [2], Engelbert A. Nonterah [5], Shane A. Norris [2], Hermann Sorgho[3], Halidou Tinto[3],
Stephen Tollman [8], Sarah E. Graham [11], Cristen J. Willer [11,12,13], AWI-Gen study*, H3Africa Consortium*,
Scott Hazelhurst [1,14] & Michèle Ramsay [1,15✉]

Genetic associations for lipid traits have identified hundreds of variants with clear differences across European, Asian and African studies. Based on a sub-Saharan-African GWAS for lipid traits in the population cross-sectional AWI-Gen cohort ($N = 10,603$) we report a novel LDL-C association in the *GATB* region (*P*-value=$1.56 \times 10^{-8}$). Meta-analysis with four other African cohorts ($N = 23,718$) provides supporting evidence for the LDL-C association with the *GATB/FHIP1A* region and identifies a novel triglyceride association signal close to the *FHIT* gene (*P*-value =$2.66 \times 10^{-8}$). Our data enable fine-mapping of several well-known lipid-trait loci including *LDLR, PMFBP1* and *LPA*. The transferability of signals detected in two large global studies (GLGC and PAGE) consistently improves with an increase in the size of the African replication cohort. Polygenic risk score analysis shows increased predictive accuracy for LDL-C levels with the narrowing of genetic distance between the discovery dataset and our cohort. Novel discovery is enhanced with the inclusion of African data.

[1] Sydney Brenner Institute for Molecular Bioscience, Faculty of Health Sciences, University of the Witwatersrand, Johannesburg, South Africa. [2] South African Medical Research Council/University of the Witwatersrand Developmental Pathways for Health Research Unit, Department of Paediatrics, School of Clinical Medicine, Faculty of Health Sciences, University of the Witwatersrand, Johannesburg, South Africa. [3] Clinical Research Unit of Nanoro, Institut de Recherche en Sciences de la Santè, Nanoro, Burkina Faso. [4] Department of Chemical Pathology, National Health Laboratory Service, Faculty of Health Sciences, University of the Witwatersrand, Johannesburg, South Africa. [5] Navrongo Health Research Centre, Ghana Health Service, Navrongo, Ghana. [6] C.K. Tedam University of Technology and Applied Sciences, Navrongo, Ghana. [7] African Population and Health Research Center, Nairobi, Kenya. [8] MRC/Wits Rural Public Health and Health Transitions Research Unit (Agincourt), School of Public Health, Faculty of Health Sciences, University of the Witwatersrand, Johannesburg, South Africa. [9] Department of Public Health, School of Health Care Sciences, Faculty of Health Sciences, University of Limpopo, Polokwane, South Africa. [10] Department of Pathology and Medical Sciences, School of Health Care Sciences, Faculty of Health Sciences, University of Limpopo, Polokwane, South Africa. [11] Department of Internal Medicine, Division of Cardiology, University of Michigan, Ann Arbor, MI 48109, USA. [12] Department of Computational Medicine and Bioinformatics, University of Michigan, Ann Arbor, MI 48109, USA. [13] Department of Human Genetics, University of Michigan, Ann Arbor, MI 48019, USA. [14] School of Electrical and Information Engineering, University of the Witwatersrand, Johannesburg, South Africa. [15] Division of Human Genetics, National Health Laboratory Service and School of Pathology, Faculty of Health Sciences, University of the Witwatersrand, Johannesburg, South Africa. [16]These authors contributed equally: Ananyo Choudhury, Jean-Tristan Brandenburg *Lists of authors and their affiliations appear at the end of the paper. ✉email: Ananyo.Choudhury@wits.ac.za; Michele.Ramsay@wits.ac.za

Circulating lipid levels and their genetic associations are important indicators of the risk for developing cardio-metabolic diseases including stroke, coronary artery disease, hypertension, and are also associated with kidney disease[1–4]. Although modulated by environmental and behavioural factors, the heritability of total cholesterol (TC), low-density lipoprotein cholesterol (LDL-C), high-density lipoprotein cholesterol (HDL-C) and triglycerides (TG) has been estimated at between 0.35 and 0.76[5,6]. However, trait heritability demonstrates considerable inter-ethnic differences, some of which could be explained by environmental differences such as diet and availability of medications[7]. Although large representative datasets are sparse, at a population level lipid profiles from African Americans generally reflect lower TG, and higher HDL-C levels compared to populations from Asia and Europe[8].

Genetic associations with lipid levels have been extensively studied using candidate loci[9], SNP arrays enriched for known associated genomic regions (Metabochip)[10,11], exome SNP arrays[12–14], genome-wide SNP arrays[15,16], whole exome sequencing (WES) and whole genome sequencing (WGS)[17] approaches to reveal over 400 robustly associated independent loci. Many of the studies included multiple ethnicities mainly representing Americans of European- and African-ancestry, Hispanics[17], American Indians[18] and Asian populations[19,20]. These studies have generally shown that the majority of lipid-trait associations are universal; however, several studies also revealed population- or ancestry-specific lipid-associated loci and allelic heterogeneity[21]. A recent systematic assessment of the transferability of an European genome-wide association study (GWAS) derived signals across ancestral groups highlighted these differences, especially in Africans, despite the shared genetic architecture for lipid traits[22].

There are demonstrable benefits in using ancestrally diverse populations in genetic-association studies. These include refinement of previously associated loci by highlighting different effect sizes across ancestral populations, enhancing the potential to identify functional variants using linkage disequilibrium (LD)-based fine mapping, and the identification of associated variants that are ancestry-specific for either novel or known loci. Arguably, studies in African populations have the potential to contribute much benefit, as these populations have increased diversity, lower LD and higher population structure, but require the use of African-centric arrays and appropriate imputation panels to maximize novel discovery[23]. However, understanding the genomic architecture of lipid traits in Africans is currently dominated by studies on admixed African-ancestry individuals resident in the USA, who represent only a portion of the genetic diversity that exists on the continent. Moreover, these studies have mostly used SNP arrays based on common genetic variation in European and admixed populations[24,25]. Only a few studies on lipid traits[7,9,26,27] have been solely or predominantly based on sub-Saharan African populations. The modest sample sizes and lack of geographic spread of participants have limited our understanding of the genetics of lipid traits in these populations.

Genetic associations from a GWAS are often used as a collective set for the generation of polygenic risk score (PRS) models. The application of these models to lipid traits shows an overall high predictability. For example, in a study of 94,674 ancestrally diverse treatment-naive participants from the Kaiser Permanente members, a GWAS using electronic health records revealed novel and sex-specific genetic associations and demonstrated that a 477 SNP PRS could predict age at first use of lipid-lowering drugs with relative accuracy[28]. However, the predictive ability of these scores not only varies widely between traits but also between GWAS for the same trait due to factors such as sample size and LD architecture[29,30]. Moreover, PRS models for several traits have shown that their application to ancestries other than the ones

in the original GWAS often leads to a loss in predictive accuracy[29–33]. Accordingly, an African population was shown to have the lowest predictive accuracy using a European-derived PRS model for lipid traits[22].

In this study, we report on the use of the H3Africa SNP genotyping array (https://www.h3abionet.org/h3africa-chip), enriched for common variants in sub-Saharan Africans, to perform GWAS for fasting serum TC, LDL-C, HDL-C, and TG in the AWI-Gen cohort based on 10,603 adults, resident in six study sites across four countries in sub-Saharan Africa[34,35]. This was followed by a meta-analysis with previously published summary statistics from four sub-Saharan African cohorts reaching a combined sample size of about 24,000. We then performed an in depth assessment of the replication of previously associated loci (using the summary statistics from the Population Architecture using Genomics and Epidemiology (PAGE)[36] and Global Lipid Genetics Consortium (GLGC)[15] in AWI-Gen and other sub-Saharan African cohorts. Finally, we examined the predictability of PRS models based on GWAS from European (GLGC)[15], multi-ancestry (PAGE)[36] and sub-Saharan African populations[7], in our cohort.

## Results

The distribution of the four lipid traits (TC, LDL-C, HDL-C and TG) and associated covariates (age and sex) in the six AWI-Gen study sites are summarized in Supplementary Data 1 and Supplementary Fig. 1. The discovery GWAS for these four traits was conducted in two stages (Fig. 1). Stage 1 was association testing using the full AWI-Gen dataset ($N = 10,603$). This dataset was imputed using the African Reference Panel at the Sanger Imputation facility and only SNPs with MAF > 0.01 and Info Score>0.6 were included in the analysis. Stage 2 (building on the outcome of Stage 1) was the meta-analysis of these results with published summary statistics from four cohorts (the Uganda Genome Resource (UGR) study, the Africa-America Diabetes Mellitus (AADM) study, the Durban Diabetes Study (DDS), the Durban Case Control (DCC) study) all of which were included in the Gurdasani et al.[7] study (Fig. 1). To account for the difference in total number of variants in the two datasets (~14 M SNPs in AWI-Gen in comparison to ~22 M in Gurdasani et al.[7]), the Stage 2 analysis was restricted to the ~14M SNPs that were present in the AWI-Gen study. This difference, despite the use of the same imputation panel, was due to a more stringent MAF cut-off (MAF > 0.01) compared to that used in Gurdasani et al.[7] (MAF > 0.005).

**Adjusting for the impact of population structure.** In alignment with the geographic spread of our study sites in Eastern, Western and Southern Africa (Fig. 1), we observed a clear population structure (Supplementary Fig. 2a). Using an approach similar to the PAGE study[36], we defined the number of relevant principal components (PCs) based on a joint plot of the first twenty PCs for representative ethnolinguistic groups (Supplementary Fig. 2b). Stage 1 GWAS was performed using a linear mixed model-based approach (BOLT-LMM)[37] after adjusting for age, sex, and relevant PCs. Genomic inflation scores and quantile-quantile plots (Supplementary Fig. 3) did not indicate major inflation in any of the analyses. All the results for Stage 1 GWAS are based on this approach. As an alternative approach, for each trait we also performed three independent GWASs based on participants from Eastern, Western and Southern Africa followed by meta-analysis of the results from these geographic-region-specific GWASs. Comparison of the genome-wide significant associations detected by the two approaches not only show an almost complete overlap at the

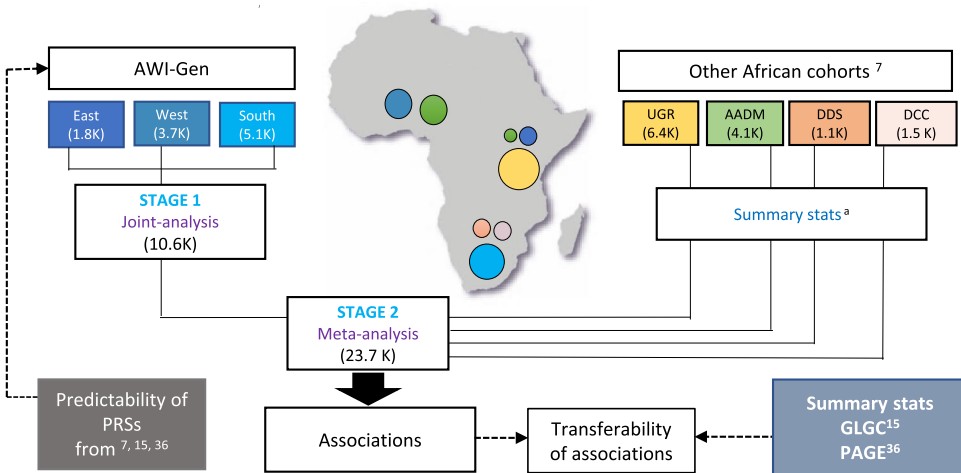

**Fig. 1 Summary of the datasets and analyses.** For each lipid trait, Stage 1 of the GWAS involved a joint analysis of the full AWI-Gen dataset (10,603 participants from Eastern, Western and Southern Africa) and Stage 2 ($N = 23,718$) involved a meta-analysis of the Stage 1 results with the summary statistics from four African cohorts included in the Gurdasani et al. 2019[7] study (Uganda Genome Resource (UGR) study, the Africa-America Diabetes Mellitus (AADM) study, the Durban Diabetes Study (DDS), the Durban Case Control (DCC) study). Approximate geographic location and sample size of the cohorts are represented by position and size of the circles in the map. Each cohort is shown in a unique colour. Assessment of transferability of associations detected in the Global Lipid Genetics Consortium (GLGC)[15] study and the Population Architecture using Genomics and Epidemiology (PAGE)[36] study was performed in various replication sets. Predictability of polygenic risk score models based on Gurdasani et al. 2019[7], GLGC[15] and PAGE[36] was assessed in the AWI-Gen dataset. The map was created using R (https://www.r-project.org/).

**Table 1 Summary of associations detected in the Stage 1 GWAS for the four lipid traits.**

| SNP | Unique ID | Beta | *P*-value (1) | *P*-value (2) | Gene/Nearby gene | Previous reports |
|---|---|---|---|---|---|---|
| *LDL-C* | | | | | | |
| rs28362286 | 1:55529215:C:A | 0.942 | 3.70E-73 | 7.74E−71 | PCSK9 | Yes |
| rs12740374 | 1:109817590:G:T | 0.125 | 3.40E-15 | 1.32E−15 | CELSR2 | Yes |
| rs2435386 | 2:21414760:C:T | −0.079 | 1.80E-08 | 1.22E−08 | RP11-79J24.1, TDRD15 | No_A |
| rs35804417 | 4:152601951:G:A | −0.174 | 4.10E-08 | 1.56E−08 | GATB | No |
| rs75143493 | 6:160946747:T:G | −0.327 | 5.90E-10 | 1.29E−09 | LPA, LPAL2 | No_B |
| rs73015020 | 19:11192550:G:A | 0.136 | 1.40E-17 | 5.10E−17 | LDLR, SMARCA4 | Yes |
| rs7412 | 19:45412079:C:T | 0.453 | 2.70E-117 | 2.24E−118 | APOE | Yes |
| *HDL-C* | | | | | | |
| rs2070895 | 15:58723939:G:A | −0.103 | 2.00E-13 | 7.63E−14a | ALDH1A2/ LIPC | Yes |
| rs34065661 | 16:56995935:C:G | −0.324 | 1.60E-36 | 2.05E−37 | CETP | Yes |
| *TG* | | | | | | |
| rs326 | 8:19819439:A:G | 0.096 | 2.40E-11 | 5.05E−12 | LPL | Yes |
| rs2070895 | 15:58723939:G:A | −0.083 | 2.70E-09 | 2.87E−09 | ALDH1A2/ LIPC | Yes |
| rs12721054 | 19:45422587:A:G | 0.214 | 4.00E-24 | 1.74E−22 | APOC1 | Yes |
| rs114139997 | 21:46875775:G:A | 0.235 | 2.10E-08 | 3.70E−09 | COL18A1 | Yes |
| *TC* | | | | | | |
| rs28362286 | 1:55529215:C:A | 0.794 | 1.70E-52 | 9.38E−51 | PCSK9 | Yes |
| rs12740374 | 1:109817590:G:T | 0.100 | 2.20E-10 | 2.21E−11 | CELSR2 | Yes |
| rs73015020 | 19:11192550:G:A | 0.112 | 2.00E-12 | 4.47E−12 | LDLR SMARCA4 | Yes |
| rs7412 | 19:45412079:C:T | 0.323 | 1.10E-60 | 3.22E−60 | APOE | Yes |

LDL-C: low-density lipoprotein cholesterol.
HDL-C: high-density lipoprotein cholesterol.
TG: triglycerides.
TC: total cholesterol.
Unique ID: Summarizes chromosome, position and the alleles.
*P*-value (1): *P*-value (two-tailed, not adjusted for multiple comparisons, calculated using BOLT-LMM)) for the joint analysis of the full AWI-Gen dataset.
*P*-value (2): *P*-value (two-tailed not adjusted for multiple comparisons, estimated using METASOFT) for the meta-analysis of East, West and South African subsets.
Previous reports: Association of the SNP with the same or related traits according to GWAS catalog and literature.
No_A: SNP has no known association with the trait but has been shown previously to be associated with one or more related traits.
No_B: Although no report for SNP, the corresponding gene contains one or more SNPs associated with the trait.
aPeak for the region was observed in a neighbouring SNP-rs1800588 (*P*-value = 7.63E−14).

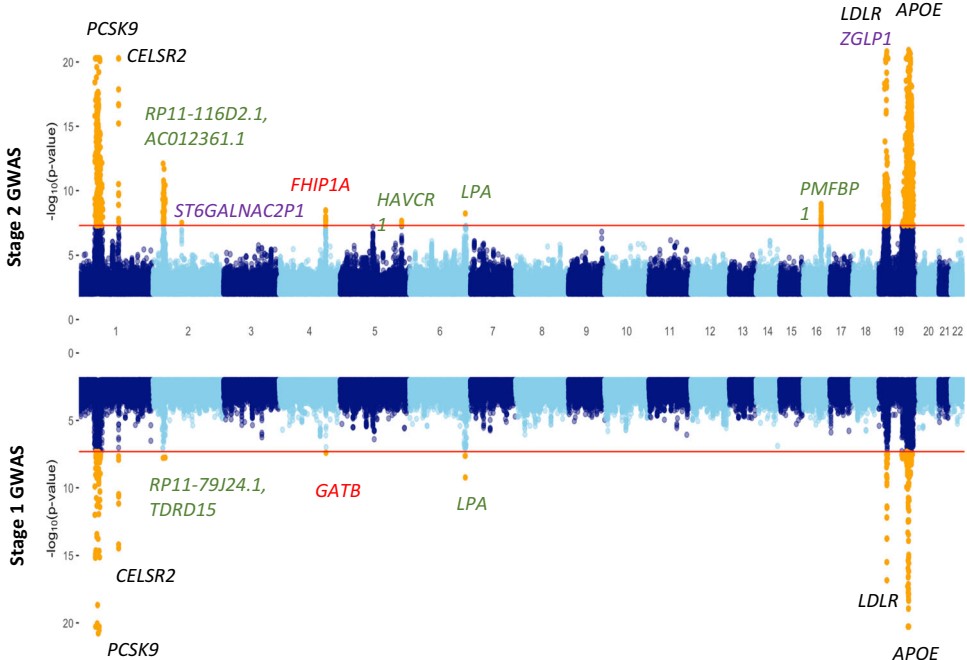

**Fig. 2 Genome-wide associations for LDL-C.** Miami plot showing summary data for Stage 1 GWAS (AWI-Gen, downward facing, $N = 10,603$) and Stage 2 GWAS (meta-analysis of AWI-Gen and four African cohorts, upward facing, $N = 23,718$). $P$-values (two-tailed, not adjusted for multiple comparisons, calculated using BOLT-LMM for Stage 1 GWAS and METASOFT for Stage 2 GWAS) are truncated at $10^{-20}$ for clarity. The red horizontal lines show the genome-wide significance threshold ($5 \times 10^{-8}$) and SNPs with $P$-values below this threshold are shown in orange. The loci corresponding to the region showing novel association in Stage 1 and Stage 2 GWAS are indicated in red. Other possible novel loci that reached genome-wide significance only in the Stage 2 analysis are shown in purple. Known LDL-C associated regions that were represented by novel lead SNPs are shown in green. Loci represented by lead SNPs that are well-known for LDL-C associations across multiple studies, including ours, are shown in black.

genomic locus level but also show very similar $P$-value estimates for most of the lead SNPs (Table 1, Supplementary Data 2).

**Genome-wide associations in the AWI-Gen cohort.** The Stage 1 GWAS identified 12 independent (characterized using FUMA[38]) genomic regions that were associated with at least one of the four lipid traits at a genome-wide significance threshold of $P$-value $<5 \times 10^{-8}$ (Fig. 2, Supplementary Figs. 4–6; Table 1; Supplementary Data 3), with 11 of the 12 signals mapping to well-known lipid-associated regions in/near genes such as PCSK9, APOE, CELSR2 and LDLR. A novel genome-wide significant signal for LDL-C, rs35804417, mapped to an intron of the GATB gene on chromosome 4 (Fig. 2, Table 1). The signal also emerged as a suggestive signal in the GWAS for TC (Supplementary Data 4). While the absence of any previous signal for this SNP in the GWAS literature could be due to its near absence in non-African populations (minor allele frequency (MAF) for African populations = 0.06; MAF for European population = 0 in the 1000 Genomes dataset[39]; MAF = 0.04 in African Americans and MAF < 0.0002 for non-African populations in the gnomAD database[40]), the lack of any previous lipid signal in the extended genomic region (+/−500 kb) around this SNP (investigated using GWAS catalog[41] and PhenoScanner[42,43] suggests that this association is ancestry/continent-specific.

The previously detected association of the LPA locus with LDL-C was represented by a novel lead SNP (rs75143493) that also lacked any report of previous association either in GWAS catalog or PhenoScanner[41–43] (Table 1). This SNP also has a very low minor allele frequency in non-Africans (Absent in 1000 Genomes European and Asian populations;[39] MAF = 0.015 in African Americans and MAF < 0.001 in non-African populations in the gnomAD database[40]) which could explain its absence in previous GWASs. Similarly, the lead SNP rs2435386 near the

well-known TDRD15 gene also has no reported associations in the GWAS catalog[41]. However, a PhenoScanner search shows this SNP to be associated with high cholesterol as well as cholesterol lowering medication. The geographic-region-specific analyses (Supplementary Data 5) identified a signal (rs12721096) in the well-known lipid-associated locus APOC3 to be significantly associated with TG in the GWAS of the Southern African participants. Similarly, several SNPs in the SENP7 gene on chromosome 3, with the lead SNP rs4683845, showed association with TG only in the West African GWAS.

**Fine mapping of lipid-trait loci.** We used three different tools, FINEMAP[44], PAINTOR[45] and CAVIAR[46] to identify 95% credible sets for the genome-wide significant loci and also to narrow down causal variants, where possible. The fine mapping for PCSK9 in the AWI-Gen dataset, performed using FINEMAP with stepwise conditioning, identified rs28362286 (PCSK9 p.Cys679Ter) as a potential causal variant for both LDL-C and TC. The prediction was supported by the other two methods. This stop-gained variant is nearly absent in non-Africans in the 1000 Genomes[39] and GnomAD datasets[40] and consequently has been detected as associations in only three of the previous lipid-trait GWAS that were either African based[7] or included a substantial portion of African ancestry[36,47]. rs28362286 was observed as the lead variant for associations with TC and LDL-C in both Stage 1 and Stage 2 GWAS (Supplementary Fig. 7). In addition, we observed a marked reduction of credible set size of the signal near the well-known LDLR locus in the African dataset (Fig. 3a) in comparison to an European GWAS[48] of comparable size (Fig. 3a). Although the top SNPs from a locus often differed between the two datasets, several other well-known lipid loci such as LPAL2, LIPC, CELSR2 and APOC1 showed a narrower peak in

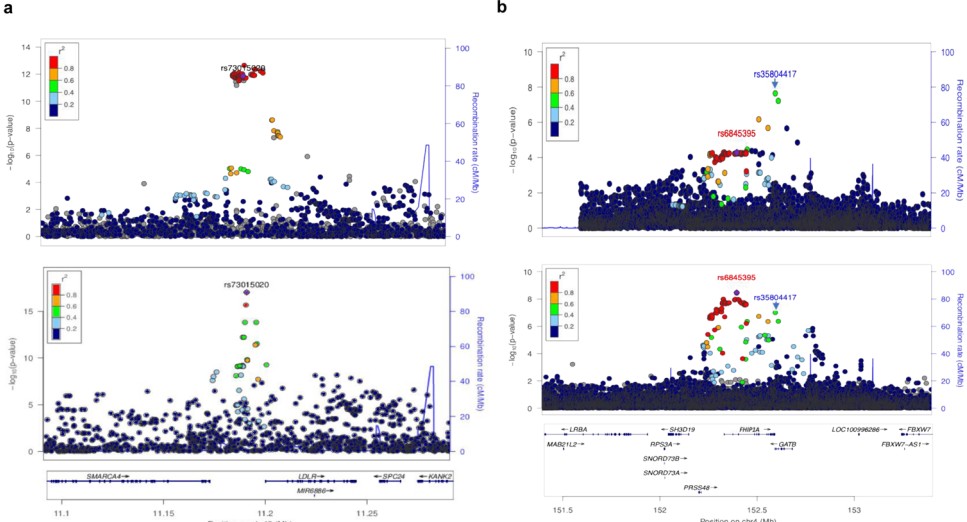

**Fig. 3 Fine mapping and novel association for LDL-C. a** Locuszoom (http://locuszoom.org/) plots showing LDL-C association around the *LDLR* region. The plot on top is based on the Prins et al. 2017[48] study ($N = 9961$). The 95% credible set in this study included over 40 SNPs (inferred using FINEMAP based results from CausalDB (http://mulinlab.org/causaldb/)). The bottom plot shows the same region in the AWI-Gen Stage 1 GWAS ($N = 10,603$). Here, the region is represented by a much narrower peak and the 95% credible set, inferred using FINEMAP, includes only two SNPs. **b** Locuszoom plot showing associations around the *GATB* region in the Stage 1 (top) ($N = 10,603$) and Stage 2 (bottom) ($N = 23,718$) analysis. Although, the lead SNP from Stage 1 GWAS, rs35804417 (pointed to by blue arrow), missed genome-wide significance, a set of SNPs, with the lead SNP rs6845395 (purple diamond), from the neighbouring *FH1P1A* gene was found to be significant in the Stage 2 GWAS.

the AWI-Gen GWAS in comparison to the European GWAS study (Supplementary Figs. 8–11).

**Meta-analysis with other African cohorts**. Stage 2 of our GWAS, based on a meta-analysis of the results from the Stage 1 GWAS and summary statistics for four other African cohorts, identified about 15 additional independent genomic regions associated with one or more of the four lipid traits (Table 2; Fig. 2; Supplementary Figs. 4–6 and 12; Supplementary Data 6). Most of the signals were within or near known lipid-associated genes such as *ABCA1*, *LPL* and *APOB* (Table 2). Among these, rs33918808 (in *ABCA1*) and rs2126263 (near *LPL*) were observed as suggestive hits in the Stage 1 GWAS (Supplementary Data 3). The novel lead SNP for the LDL-C association with *LPA* identified in the Stage 1 GWAS also emerged as the lead SNP in the Stage 2 GWAS (Supplementary Fig. 13). Moreover, several other associations such as *BUD13* with TG (rs116588420) and *PMFBP1* with LDL-C (rs4788609), detected in the Stage 2 GWAS, were also found to be represented by novel lead SNPs.

The novel signal in the *GATB* gene showing LDL-C association in the Stage 1 GWAS was found to be nominally replicated (P-value < 0.05) in the AADM cohort and was close to the genome-wide significance threshold ($P\text{-value} = 9.83 \times 10^{-8}$) in the Stage 2 GWAS (Fig. 3b). Moreover, the Stage 2 GWAS identified multiple signals (lead SNP rs6845395) with moderate LD (0.5) in a nearby region harbouring the *FHIP1A* (also known as *FAM160A1*) gene at the genome-wide significance threshold (Fig. 3b). Two of the other associations for LDL-C, rs141822553 near the *ST6GAL-NAC2P1* gene on chromosome 2 and rs114810281 near the *ZGLP1* gene (close to but independent of the main *LDLR* signal) on chromosome 19, are also strong candidates for being novel signals. A potential novel signal for TG was observed in an intergenic region near the *FHIT* gene (rs75064672). The comparison of results from various meta-analysis models demonstrated that most of these associations were significant at the standard genome-wide significance threshold irrespective of whether a fixed, random or binary effect model was considered (Supplementary Data 6).

To assess functional relevance of the associations we used the gene-based analysis option in MAGMA (within the FUMA toolkit)[38]. In this analysis, only the associations that localize within genes are considered and the significance is assessed on a gene-based Bonferroni P-value threshold of $5 \times 10^{-6}$. This analysis identified 43 signals in Stage 1 and 53 signals in Stage 2 GWAS. Although comparison with the GWAS catalog[41] showed that most of the genes originate in regions with previous signals, it identified novel signals in both Stage 1 and 2 GWAS. The analysis of gene set enrichment further showed that most significantly enriched gene sets were relevant to the biology of lipid metabolism (Supplementary Data 7).

**Transferability of signals from large consortium studies**. To evaluate the transferability of signals from the predominantly European-based GLGC Consortium study[15] to the African dataset, we assessed the P-value of the signals from the GLGC study in the Stage 1 GWAS, Stage 2 GWAS and the four African cohorts included in the meta-analysis. To delineate independent loci for evaluation, genome-wide significant signals in the GLGC summary statistics that were within 50 kb of each other were considered as belonging to the same locus. For each locus, if any of the SNPs that were genome-wide significant in the GLGC GWAS showed $P\text{-value} < 5 \times 10^{-4}$ in an African cohort, the locus was considered to be replicated/transferable. The cut-off was decided on the premise that each of four lipid traits were represented by about 100 independent loci in the GLGC dataset and therefore a minimum correction for 100 tests was necessary (Corrected replication P-value threshold 0.05/100). In addition, we also assessed replication at a more stringent level (genome-wide $P\text{-value} < 5 \times 10^{-8}$) and a less stringent threshold (nominal threshold P-value < 0.05). To limit the impact of SNP-set size variations among replication datasets (some included ~14 M and some ~22 M variants), the entire analysis was restricted only to the ~14 M SNPs that were present in the AWI-Gen dataset.

A previous study on lipid-trait signals[22] has shown the transferability of signals from GLGC to vary considerably with the strength of the signals; signals with $P\text{-value} < 10^{-100}$ show

**Table 2 Summary of associations detected in Stage 2 GWAS (meta-analysis of AWI-Gen and four other African cohorts). Additional details in Supplementary Data 6.**

| SNP | Unique ID | Beta | P-value | Gene/Nearby gene | Previous reports |
|---|---|---|---|---|---|
| *LDL-C* | | | | | |
| rs28362286 | 1:55529215:A:C | 0.882 | 4.43E-113 | *PCSK9* | Yes |
| rs12740374 | 1:109817590:G:T | 0.138 | 1.46E-39 | *CELSR2* | Yes |
| rs143375141 | 2:21179426:C:T | −0.105 | 7.82E-13 | *RP11-116D2.1, AC012361.1* | No_A |
| rs141822553 | 2:84387050:C:T | −0.194 | 3.09E-08 | *ST6GALNAC2P1, FUNDC2P2* | No |
| rs6845395 | 4:152403605:C:T | −0.168 | 3.29E-09 | *FHIP1A* | No |
| rs9784624 | 5:156440014:C:T | −0.067 | 2.17E-08 | *HAVCR1, TIMD4* | No_B |
| rs75143493 | 6:160946747:G:T | −0.172 | 5.83E-09 | *LPA, LPAL2* | No_B |
| rs4788609 | 16:72165986:C:T | 0.071 | 1.03E-09 | *PMFBP1* | No_B |
| rs114810281 | 19:10415812:C:T | 0.175 | 1.34E-09 | *ZGLP1* | No |
| rs73015020 | 19:11192550:A:G | 0.150 | 8.09E-45 | *LDLR, SMARCA4* | No_A |
| rs7412 | 19:45412079:C:T | 0.535 | 9.14E-304 | *APOE* | Yes |
| *HDL-C* | | | | | |
| rs2126263 | 8:9181611:A:G | −0.065 | 1.43E-08 | *RP11-115J16.1* | Yes |
| rs3289 | 8:19823192:C:T | 0.104 | 3.16E-11 | *LPL* | Yes |
| rs2070895 | 15:58723939:A:G | −0.076 | 1.31E-23 | *LIPC* | Yes |
| rs34065661 | 16:56995935:C:G | −0.341 | 8.46E-102 | *CETP* | Yea |
| rs2292318 | 16:67985706:C:T | −0.075 | 1.59E-08 | *SLC12A4* | Yes |
| rs3744841 | 18:47117374:A:G | −0.060 | 1.04E-08 | *LIPG* | Yes |
| rs7412 | 19:45412079:C:T | −0.147 | 9.93E-20 | *APOE* | Yes |
| *TG* | | | | | |
| rs575787792 | 1:63356272:C:T | 0.252 | 3.29E-11 | *RP4-771M4.3, ATG4C* | No |
| rs75064672 | 3:59707791:C:T | −0.187 | 2.67E-08 | *RP11-719N22.1, FHIT* | No |
| rs3289 | 8:19823192:C:T | −0.130 | 3.12E-17 | *LPL* | Yes |
| rs116588420 | 11:116629766:G:T | −0.211 | 3.63E-10 | *BUD13* | No_B |
| rs2070895 | 15:58723939:A:G | −0.078 | 1.81E-17 | *LIPC* | Yes |
| rs4783961 | 16:56994894:A:G | 0.051 | 3.89E-08 | *AC012181.1, CETP* | No_A |
| rs12721054 | 19:45422587:A:G | 0.194 | 6.80E-47 | *APOC1* | Yes |
| rs114139997 | 21:46875775:A:G | 0.230 | 4.32E-13 | *COL18A1* | Yes |
| *TC* | | | | | |
| rs28362286 | 1:55529215:A:C | 0.793 | 3.39E-88 | *PCSK9* | Yes |
| rs12740374 | 1:109817590:G:T | 0.108 | 2.95E-25 | *CELSR2* | Yes |
| rs661665 | 2:21265141:A:C | −0.072 | 3.03E-11 | *APOB* | Yes |
| rs115069429 | 6:160872648:C:T | −0.133 | 4.02E-09 | *SLC22A3* | No_B |
| rs33918808 | 9:107579632:C:G | −0.071 | 9.04E-10 | *ABCA1* | No_B |
| rs34065661 | 16:56995935:C:G | −0.099 | 2.62E-09 | *CETP* | No_A |
| rs61483465 | 16:72217018:A:C | −0.081 | 7.01E-11 | *PMFBP1, RP11-328J14.1* | Yes |
| rs12151108 | 19:11197261:A:G | 0.144 | 1.03E-30 | *LDLR, SMARCA4* | Yes |
| rs7412 | 19:45412079:C:T | 0.361 | 4.98E-142 | *APOE* | Yes |

LDL-C: low-density lipoprotein cholesterol.
HDL-C: high density lipoprotein cholesterol.
TG: triglycerides.
TC: total cholesterol.
Unique ID: Summarizes chromosome, position and the alleles.
Beta, P-value: Effect size and P-value (two-tailed, not. adjusted for multiple comparisons) calculated using RE2 model implemented in METASOFT.
Previous reports: Association of the SNP with the same or related traits according to GWAS catalog, UKBB and literature.
No_A: SNP has no known association with the trait but has been shown previously to be associated to one or more related traits.
No_B: Although no report for SNP, the corresponding gene contains one or more SNPs associated with the trait.

much higher transferability compared to signals with higher P-values. To assess this further we classified signals into three categories—very strong ($P$-value $< 10^{-100}$), strong ($10^{-100}$ $< P$-value $< 10^{-20}$) and moderate ($10^{-20} < P$-value $< 10^{-8}$) and studied the levels of transferability in each of these categories separately. The results presented in Fig. 4 and Supplementary Fig. 14 show that an increase in sample size of the African cohort generally resulted in increased transferability. We observed very strong signals to show highest transferability across traits. Similarly, strong signals showed better transferability in comparison to moderate signals. Moreover, in addition to the number of loci replicated, the $P$-values at which these replications were observed also decreased consistently with the increase in the size of the replication cohort.

A similar analysis of signals from the multi-ethnic but much smaller PAGE consortium study[36] showed the same trend along with an overall higher transferability (Fig. 4, Supplementary Fig. 14) compared to signals from the GLGC study across all lipid traits. As there were very few signals with $P$-value $< 10^{-100}$, we only categorized the signals into strong ($P$-value $< 10^{-20}$) and moderate ($10^{-20} < P$-value $< 10^{-8}$), and once again observed better replication in the former compared to the latter. Although variation in the level of transferability of signals for the four traits was not consistent across the PAGE and the GLGC study, in both cases we observed the signals for LDL-C and HDL-C to show better transferability compared to the other two traits (also noted in[22]).

**Predictability of available polygenic risk score models in the AWI-Gen cohort.** Next, we assessed the extent to which PRS derived from GLGC[15], PAGE[36] and a sub-Saharan African

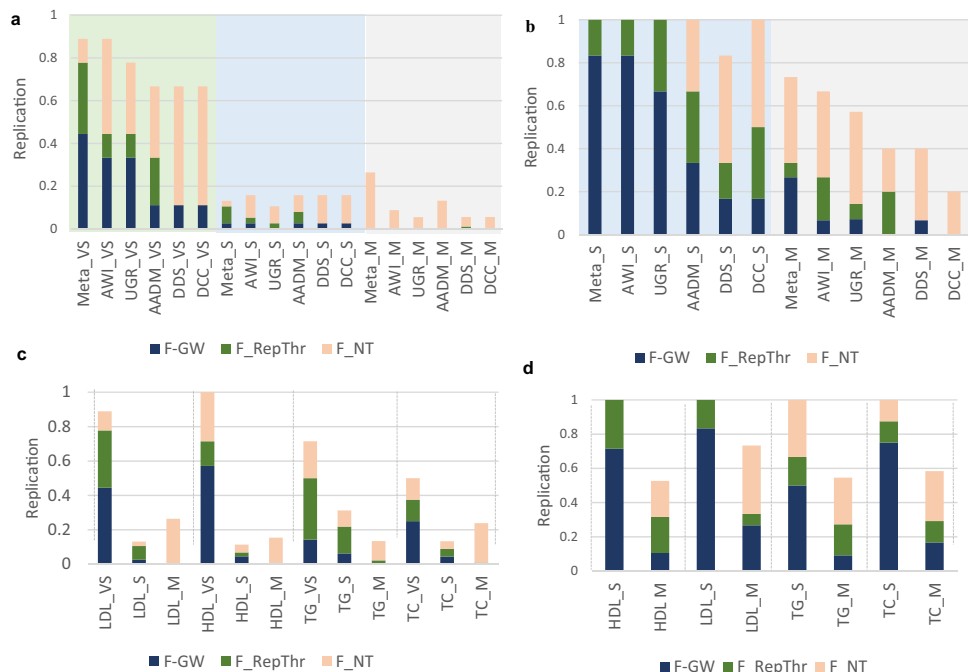

**Fig. 4 Transferability of previous lipid-trait associations to various African GWASs. a** LDL-C association signals detected in the Global Lipid Genetics Consortium (GLGC)[15] study. **b** LDL-C association signals detected in Population Architecture using Genomics and Epidemiology Consortium (PAGE)[36] study. The Y-axis shows the proportion of associated loci that are replicated in each of the individual African studies. The proportion of signals replicated at the genome-wide significance threshold P-value < $5 \times 10^{-8}$ are shown in deep blue (F-GW), at P-value < $5 \times 10^{-4}$ are shown in dark green (F_RepThr) and at the nominal threshold of P-value < 0.05 are shown in orange (F_NT). The signals from the GLGC study were partitioned on the basis of signal strength into Very Strong (P-value < $10^{-100}$) indicated by the suffix "_VS" and a green background, Strong ($10^{-20}$ > P-value > $10^{-100}$) indicated by the suffix "_S" and a blue background, and moderate ($5 \times 10^{-8}$ > P-value > $10^{-20}$) indicated by the suffix "_M" and a grey background. For the PAGE study only two categories, Strong (S) (P-value < $10^{-20}$) and Moderate (M) ($5 \times 10^{-8}$ > P-value > $10^{-20}$) were considered. The African replication datasets used in the analysis are – Stage 2 GWAS (Meta-analysis) (N = 23,718), Stage 1 GWAS (AWI-Gen) (N = 10,603), Uganda Genome Resource (UGR) study (N = 6407), Africa-America Diabetes Mellitus (AADM) study (N = 4116), Durban Diabetes study (DDS) (N = 1117) and Durban case control (DCC) study (N = 1475). Comparison of transferability of signals from **c** GLGC Consortium study, **d** PAGE Consortium study in the Stage 2 GWAS results for each of the four lipid traits are shown.

meta-analysis GWAS[7] could predict the levels of the four lipid traits in the AWI-Gen cohort participants (Fig. 5; Supplementary Data 8). The PRS derived from sub-Saharan Africans (shown as AFG) were the best predictors for LDL-C ($R^2 = 0.07$; P-value = $6.58 \times 10^{-131}$) and triglycerides ($R^2 = 0.01$; P-value = $2.02 \times 10^{-17}$) while the multi-ancestry (PAGE) based PRS better predicted HDL-C ($R^2 = 0.023$; P-value = $1.06 \times 10^{-37}$) and total cholesterol ($R^2 = 0.03$; P-value = $2.01 \times 10^{-53}$) as illustrated in Fig. 5. The sub-Saharan African models predicted better than the European model regardless of having fewer SNPs. Overall, the European-derived PRS (GLGC) had the lowest predictability for all the lipid traits in Africans. We then proceeded to evaluate the PRS stratifications for the lipid traits. Notably, the participants in the upper decile for the sub-Saharan African derived PRS had around 1.1 mmol/L higher LDL-C compared to those in the lowest decile after adjustment for age, sex and residual population structure (Fig. 5).

**Intra-continental heterogeneity in effect size and allele frequency.** Finally, to assess the level of heterogeneity between various African cohorts we compared the distribution of MAFs and effect size estimates for SNPs that were detected as lead SNPs in the Stage-2 GWAS in the AADM, UGR, DDC, DDS, AWI-Gen South African, AWI-Gen West African, AWI-Gen East African datasets (Fig. 6, Supplementary Data 9, Supplementary Figs. 15–18). Among these the UGR, AADM, AWI-Gen South African and AWI-Gen West African datasets had sample sizes ranging between 3600 and 6400 whereas AWI-Gen East African, DDC and DDS were much smaller. The comparisons revealed several trends in the variation of these two

estimates (Fig. 6; Supplementary Figs. 15–18). For example, the well-known lipid-associated SNP rs7412 (*APOE* p.Arg176Cys*) shows almost two-fold higher MAF in South African cohorts compared to both the East African cohorts and one of the West African cohorts. However, with the exception of the DDS and UGR cohorts the effect sizes for association of this SNP with LDL-C were comparable among other cohorts. The SNP rs4788609 in *PMFBP1* has similar MAFs in Eastern, Western and Southern Africa (with some intra-regional variations) but very different effect sizes (for LDL-C) in one of the Eastern and Southern African cohorts. Finally, the novel lead SNP rs6845395 for LDL-C shows major variations in both allele frequencies and effect sizes across the cohorts. Overall, the results from the three smaller cohorts DDS, DCC and AWI-Gen East showed the maximum deviation from other studies and also exhibited greater magnitude of standard error estimates (Supplementary Figs. 15–18). Although the allele frequency distributions showed geographically correlated trends in some cases, major differences within a geographic region were not uncommon (Supplementary Data 9).

**Discussion**

The lpaucity of GWAS in African populations has been consistently highlighted in recent literature[23,25,49,50]. AWI-Gen is a pan-African GWAS study (based on 6 study sites across four countries in Eastern, Western and Southern Africa) for lipid-trait genetics in sub-Saharan African populations. Before highlighting the key findings we discuss some of the strengths of the AWI-Gen study. These include the use of a single laboratory for measuring

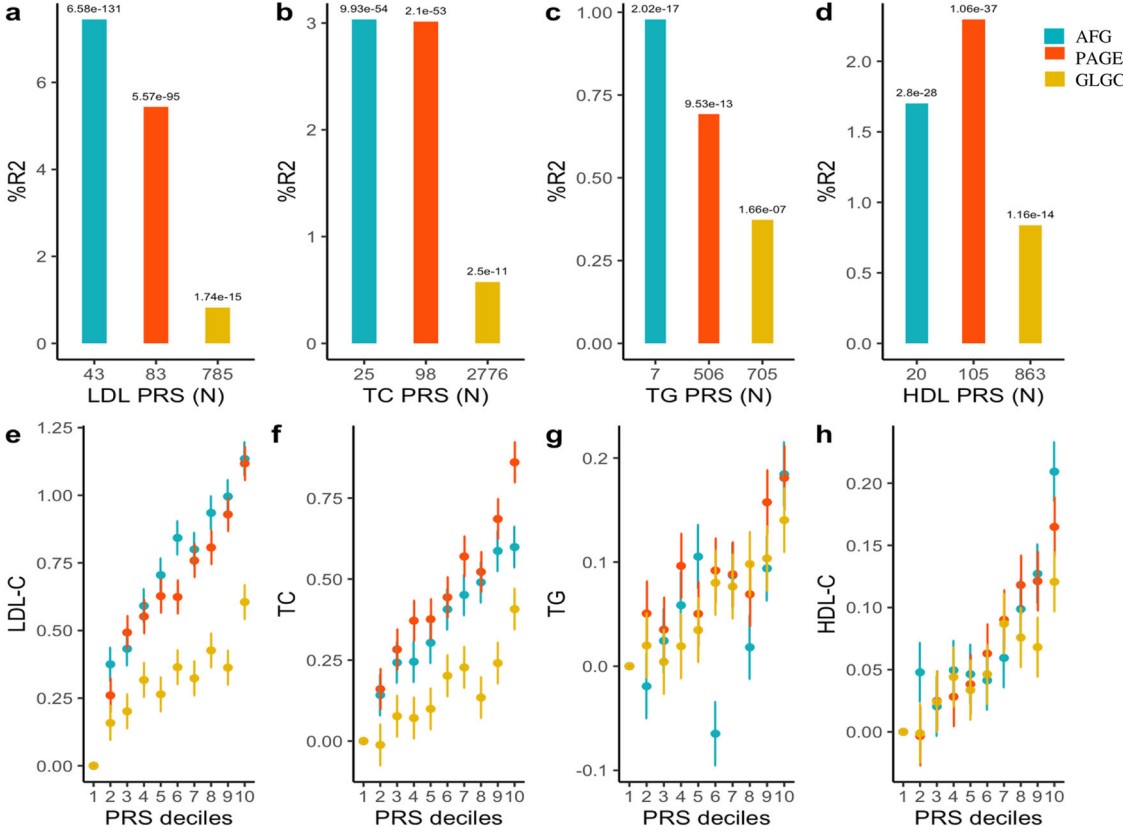

**Fig. 5 Transferability of Polygenic Risk Score (PRS) Models derived from a sub-Saharan African, an European and a Multi-ancestry GWAS to the AWI-Gen dataset.** Plots showing additional variance explained (%R2) by each PRS for (**a**) LDL-C, (**b**) TC, (**c**) TG and (**d**) HDL-C in the AWI-Gen validation dataset (N = 7103). PRS based on sub-Saharan African discovery dataset (AFG, N = 13,115 individuals)[7] is shown in blue, European (GLGC, N = 188,577 individuals)[15] in yellow and Multi-ancestry (PAGE, N = 49,839 individuals)[36] in red. Number of SNPs in each PRS is shown below and P-values (two-tailed estimates using PRSice2) are shown over the bars. All the PRS were significant for all lipid traits indicating transferability. However, the sub-Saharan African and Multi-ancestry PRS models had higher predictive accuracy compared to the European model. PRS stratification of (**e**) LDL-C, (**f**) TC, (**g**) TG and (**h**) HDL-C. Point range-plots comparing the difference in lipid-trait mean (mmol/L) of the upper PRS decile from the lowest, stratified by the discovery datasets are shown. The error bars show mean ± 95% confidence intervals. Additional details in Supplementary Data 8.

lipid biomarker levels for all the samples, as well as the same genotyping array and joint imputation for capturing genotype data which ensured much higher comparability of the data from different parts of the continent. The population cross-sectional nature of the cohort, in contrast to cohorts from disease-based studies such as AADM, DDC and DDS, was also expected to reduce the influence of other conditions on lipid levels. Finally, almost all the AWI-Gen participants were naive to medication for dyslipidemia and other cardiometabolic diseases removing another potential confounder from the study.

A key requirement for the success of a pan-African GWAS is the identification of an optimal approach to control for population structure. This is driven by the fact that in addition to pronounced genetic differences between Eastern, Western and Southern African populations, genetic differences are common even among participants from the same geographic region. For example, a recent study reported that fine-scale population structure was strong enough to influence association signals among the South African participants from our cohort[51]. This, added to environmental and lifestyle differences within and between geographic regions, as observed in several phenotype studies based on the AWI-Gen cohort[34,35,52,53], could limit the ability of modest sample size GWAS based on pan-African datasets to identify associations accurately. We have employed a LMM based approach to conduct association testing on the full dataset with principal components as covariates for the Stage

1 GWAS. Given the modest sample size of our study, we considered this composite or "mega-analysis" to be appropriate as it was expected to have higher statistical power compared to meta-analysis of separate GWAS for the three African regions[36]. However, the high concordance in the outcomes of the mega- and meta-analyses of AWI-Gen East, West and South African data demonstrates that these two approaches perform comparably in our study. However, further in-depth analysis from other similar cohorts will be required to develop comprehensive guidelines for addressing population structure and heterogeneity in pan-African GWAS based on modest sample sizes.

To bolster the power of our GWAS, we performed a meta-analysis of the Stage 1 results with the summary statistics from four African cohorts included in the Gurdasani et al. 2019 study[7]. The combined sample size of around 24,000 participants makes this study considerably larger than previous GWASs for lipid traits conducted in sub-Saharan African populations. We restricted the analysis to only those SNPs that were included in the final AWI-Gen dataset and in at least three of the four other cohorts. While this could have limited our potential for discovering novel associations, this approach ensured that the associations were based on a significant proportion of the combined sample set and are thereby relatively robust. Another ongoing discussion given the overall lower LD in African genomes, concerns the optimal P-value cut-off for African GWAS/meta-analysis, especially those using imputed data. Approaches such as using an overall lower threshold of

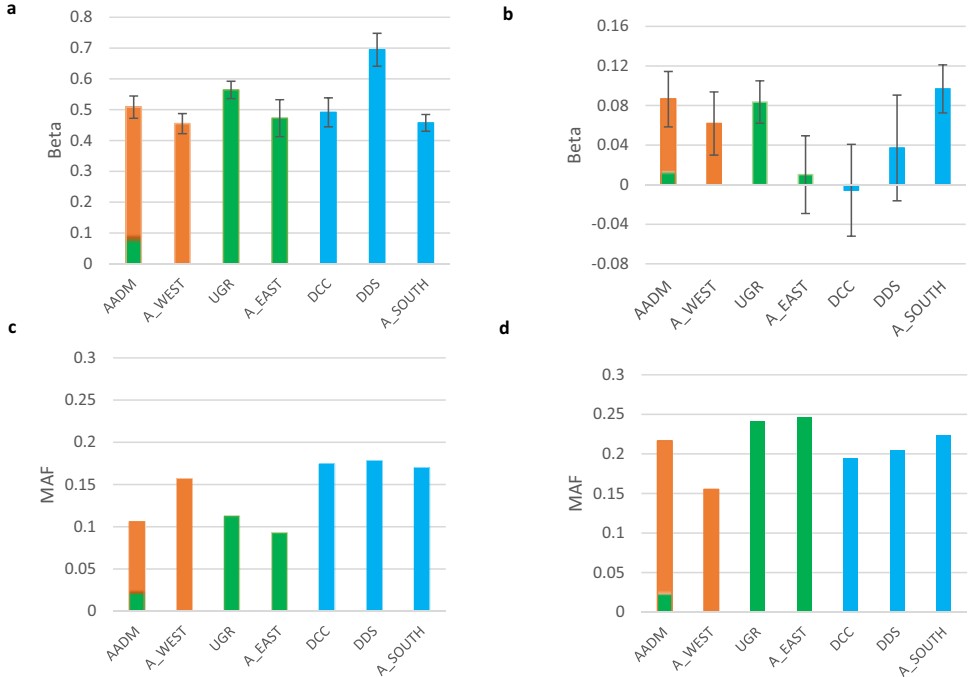

**Fig. 6 Heterogeneity of effect size and minor allele frequencies of two LDL-C association signals in African datasets.** (**a**), (**c**) show effect size and MAF for rs7412 and (**b**), (**d**) show effect size and MAF for rs4788609. The GWASs compared include AWI-Gen West African (A_WEST) GWAS (N = 3763), AWI-Gen East African (A_EAST) GWAS (N = 1755), AWI-Gen South African (A_SOUTH) GWAS (N = 5085), Uganda Genome Resource (UGR) study (N = 6407), Africa-America Diabetes Mellitus (AADM) study (N = 4116), Durban Diabetes (DDS) (N = 1117) Study and Durban Case Control (DDC) Study (N = 1475). The West African populations are shown in orange, East African populations in green and Southern African populations in blue. AADM due to inclusion of both East and West African participants is shown in two colours. Error bars show effect sizes ± standard errors.

significance ($< 5 \times 10^{-9}$)[7] or allele frequency dependent cut-offs have been recommended[36,54]. As there is a lack of consensus, we have used the currently accepted genome-wide significance threshold *P*-value$< 5 \times 10^{-8}$. However, our novel signal (rs6845395) for LDL-C in the Stage 2 GWAS reached a *P*-value of $3.28 \times 10^{-9}$.

In addition to replicating many known lipid associations, we identified one novel locus for LDL-C on chromosome 4. This signal in the Stage 1 GWAS was led by an intronic SNP (rs35804417) in the *GATB* gene (Fig. 3b). The nominal replication of this SNP in the AADM cohort as well as the observation of genome-wide significant signals in linked SNPs from the nearby *FHIP1A* gene in the Stage 2 GWAS provides strong support for association at this locus. Previous associations for trans-fatty acid levels in this genomic region further suggests a possible indirect link of this genomic region to lipid levels.

We also detected a novel association for TG in the Stage 2 GWAS. The lead TG-associated SNP near *FHIT* from the Stage 2 meta-analysis (rs75064672), reached a *P*-value < 0.05 in three independent African cohorts suggesting that this is a reliable association signal. Several variants from this gene have been detected to be associated with body composition related traits including body mass index, obesity and lean mass[41].

The extreme rarity of the variant alleles for these novel associations in European populations did not allow for the replication of these signals in some of the global cohorts. Moreover, the look-up of these two novel signals in the African ancestry GLGC cohort (~86 K samples) as well as the trans-ethnic meta-analysis of the GLGC cohort (~1.5 M samples) from the recent GLGC study[55] showed that these SNPs occur at much lower frequencies in both these cohorts in comparison to AWI-Gen (Supplementary Data 10). The absence of replication in this GLGC study suggests these signals are continent specific or involve gene-environment

interactions that are not applicable to African-Americans and other populations. The non-replication of the TG signal could also have been impacted by the fact that the summary statistics from this GLGC study are based on log(TG).

In addition to signals at the continental level we report a novel West African specific signal for association with TG in the *SENP7* gene. Despite being a globally common SNP, this lead SNP has no previous report for association with any lipid traits; however, a PhenoScanner[42,43] search showed another SNP from the region, rs149851118, to be associated (*P*-value $= 1.93 \times 10^{-8}$) with "Treatment with lipantil micro 67 mg capsule" in a UK Biobank study. Fenofibrate, the active ingredient of the drug, is widely used to treat abnormal lipid levels, which suggests a possible involvement of the variant/region in modulating lipid metabolism. However, further investigations would be required to establish a functional connection between this genomic region and lipid traits.

The absence of large-effect novel associations in the meta-analysis probably hints at saturation in the discovery of major lipid traits associated with globally common variants. However, the detection of novel African-specific lead SNPs, potential causal variants and relatively smaller credible sets for some well-known lipid-associated loci highlights the importance of including large African GWAS in fine-mapping of lipid traits.

To assess the extent to which current genetic knowledge gained from GWAS conducted in European populations could benefit risk prediction in populations from other continents, it is critical to ascertain that the same loci are associated with the trait across such populations. An in-depth study of the transferability of known lipid-trait associations detected in a predominantly European cohort, to GWAS conducted in East Asian and African ancestry populations has shown that of the hundreds of loci associated with lipid traits, only a portion of signals replicate in

African populations even at a moderate replication threshold of $P$-value $< 5 \times 10^{-3}$[22]. The relatively modest size of the African cohort (~6400 samples) included in that study could have been a major reason for the low transferability of signals to African GWAS. To test whether the use of larger African replication cohorts could address this lack of transferability, we studied the transferability of signals from two large discovery cohorts (GLGC[15] study and PAGE[36] study) to various African replication sets (four cohorts from the Gurdasani et al., 2019 study, Stage 1 GWAS and the Stage 2 GWAS). Signals from smaller discovery GWAS, due to higher effect sizes can be expected to be more transferable compared to lower effect signals in larger studies. Consistent with the differences in GWAS sizes, we observed that the signals from the PAGE study showed considerably better transferability compared to the much larger GLGC study. Moreover, the multi-ethnic nature of the PAGE study cohort compared to predominantly European samples in the GLGC study could have contributed to these differences. Nevertheless, for both the discovery cohorts, the transferability was found to increase more or less consistently with the size of the replication cohort and in general the SNPs with stronger signals (lower $P$-values) from a study showed better transferability in comparison to modest signals from the same study. In line with the previous observation[22] that transferability of signals is not uniform across traits and associations for some traits (such as HDL-C) have much higher transferability in comparison to others (such as TG), for both the discovery sets and across all the replication cohorts, we observed that LDL-C and HDL-C showed higher transferability in comparison to TG.

The promise of global application of precision medicine approaches will require efficient transferability of polygenic risk score (PRS) models across diverse populations. However, since research is currently largely based on European populations, there is a strong bias for potential beneficiaries. Studies on the transferability/predictability of existing PRS models for various traits from the European-based discovery cohorts to East Asian and African populations are of critical importance[29–33]. Several of these studies have demonstrated an overall poor transferability of existing European-based PRS models to African ancestry populations for most of the traits investigated. Allele frequency and LD differences as well as environmental differences have been suggested as the major sources of the low transferability of PRS models to Africans. The study by Kuchenbaekar and colleagues[22] highlighted the lack of transferability of European-based PRS models for lipid traits to African populations. The analysis of predictability of PRS based on summary statistics from three different studies (Gurdasani et al., 2019 study which is African ancestry, GLGC study which is European ancestry and PAGE study which is multi-ancestry) in the AWI-Gen cohort, showed progressively better predictability for signals from genetically closer cohorts for the lipids traits such as LDL-C. It is therefore critical to increase the number of non-European participants in GWAS to enhance polygenic prediction across diverse populations[56].

As factors such as allele frequency and effect size could strongly influence the detection of a signal in a GWAS, we performed an in-depth comparison of effect sizes and MAFs of association signals detected in our study in the Eastern, Western and Southern African subsets of our cohort and four other African cohorts. The inclusion of at least two cohorts from each of three African regions (AWI-Gen South Africa, DDS and DDC representing Southern Africa, UGR and AWI-Gen East representing Eastern Africa and AADM and AWI-Gen West Africa representing Western Africa) enabled us to investigate if there are systematic geographic trends in these estimates. Although our results show considerable heterogeneity between African cohorts, we did not see clear geographic differences suggesting that there is

probably more intra-region variation compared to inter-region variation in the effect size and frequencies of these SNPs in the cohorts studied. The high level of heterogeneity also indicates that effect sizes for a SNP detected in one particular African cohort might not correspond to or have predictive relevance for other African cohorts, including those from the same geographic region. However, it needs to be highlighted that due to factors such as differences in sample size, age, proportion of participants with type 2 diabetes (as some of these were diabetes-based cohorts and type 2 diabetes might have influenced lipid levels) that exists between them, these cohorts are not completely comparable. The use of more homogenous cohorts across the continent will be required to perform a robust investigation into possible geographic trends.

The modest sample size of the AWI-Gen study is a limitation to the power of the GWAS. In addition, pronounced differences in genetic variation, and environmental and lifestyle factors between the AWI-Gen study sites may have impacted the overall statistical power. Despite these challenges, the detection of novel signals and novel lead SNPs for lipid traits in the AWI-Gen cohort and the meta-analysis with other African studies emphasise the promise and potential of African GWAS in enhancing our understanding of the genetic aetiology of lipid traits and dyslipidemia. In addition to demonstrating that the extent of transferability might vary widely between lipid traits and discovery cohorts, our results suggest that the transferability for PRS might be further enhanced with the use of more multi-ethnic based GWAS datasets.

## Methods

**Ethics**. This study was approved by the Human Research Ethics Committee (Medical) of the University of the Witwatersrand (Wits) (protocol numbers M121029 and M170880). In addition, each research site obtained approval from their local ethics review board prior to commencing any participant-related activities.

**Study cohort**. The participants in this study are part of the Africa Wits-INDEPTH partnership for Genomics studies (AWI-Gen) which aims to examine genetic and environmental factors related to cardiometabolic diseases in Africans. It is part of the Human Heredity and Health in Africa Consortium (H3Africa). From 2012 to 2016, ~12,000 participants, primarily between the ages of 40 to 60 years were enrolled across the six AWI-Gen centres, and a further 552 over the age of 60 enrolled at the Agincourt centre. Following community engagement and individual informed consent, data and samples were collected from six study centres in four SSA countries: in South Africa, the MRC/Wits Agincourt Health and Demographic Surveillance System Site (HDSS) (referred to as Agincourt), the Dikgale HDSS of the University of Limpopo, and the Soweto centre which is coordinated by the South African Medical Research Council/Wits Developmental Pathways for Health Research Unit (DPHRU); in Kenya, the African Population and Health Research Center HDSS in Nairobi; in Ghana, the Navrongo HDSS in the Navrongo Health Research Centre; and in Burkina Faso the Nanoro HDSS hosted by the Institut de Recherche en Sciences de la Santé Clinical Research Unit. Participants were given a small compensation to cover their travel and incidentals. Further details are available in Ramsay et al.[34] and Ali et al.[35].

**Lipid measurements**. Fasting serum lipids were analyzed using a Randox Plus clinical chemistry analyzer (Crumlin, Northern Ireland, UK) using colorimetric assays and all assays were performed at the DPHRU laboratories in Soweto, Johannesburg. The concentrations for triglycerides (TG), total cholesterol (TC), high density lipoprotein cholesterol (HDL-C) were determined directly from the assay. The concentration of low-density lipoprotein cholesterol (LDL-C) calculated using the Friedewald equation[57]. The coefficient of variation of the laboratory measurements for lipids was less than 1.5 %. Each lipid phenotype was checked for extreme outliers and inverse normal transformed prior to genetic-association analysis.

**Genotyping and pre-imputation QC**. Approximately 11,000 samples were genotyped on the 2.3 M SNP H3Africa array at Illumina® FastTrack™ Microarray services (Illumina, San Diego, USA). The genotype calling was performed using the Illumina pipeline. Pre-imputation quality control (QC), performed using H3ABioNet/H3Agwas pipeline (https://github.com/h3abionet/h3agwas) and involved the removal of SNPs showing missingness greater than 0.05, MAF less than 0.01, and Hardy-Weinberg equilibrium (HWE) $P$-value less than 0.0001. In addition, duplicates, X chromosome, Y chromosome and mitochondrial SNPs, and SNPs that did not match the GRCh37 references alleles were also removed. Samples which were potential duplicates (PIHAT > 0.9), had a missing SNP genotyping rate

greater than 0.05, and showed sex inconsistencies (between recorded and genetic sex) were excluded. Population stratification in this dataset, was assessed using a principal component (PC) analysis, based on a LD pruned subset of SNPs, using the smartPCA program implemented in EIGENSTRAT[58]. As the genotyping was performed in 4 independent batches, with some overlapping samples, both PC and genotype concordance of these overlapping samples was used to identify possible batch effects. Both the approaches ruled out any serious batch effect in the dataset.

**Imputation and pre-analysis QC.** Imputation was performed using the African Genome Resources reference panel (EAGLE2 + PBWT pipeline) at the Sanger Imputation Server (https://imputation.sanger.ac.uk/). Post imputation quality control (QC) involved removal of indels, rare SNPs (with minor allele frequency (MAF) < = 0.01) and poorly imputed SNPs (Info score < = 0.6) resulting in the final dataset containing 10,603 participants and 13.98 M SNPs.

**Stage 1: Association analysis.** Using residuals adjusted for age, sex and first $N$ PCs ($N = 8$ for full datasets, $N <= 4$ for East, West and South African specific analyses) and imputed data using dosage format, we performed an association analysis using BOLT-LMM implemented in the h3agwas pipeline (github.com/ h3abionet/h3agwas/assoc/assoc.nf)[59]. For each dataset we employed BOLT-LMM[37] with leave-one-chromosome-out (LOCO) analysis and 1,000,000 SNPs to build models. Independent SNPs were pruned using PLINK[60] ("--indep-pairwise" option with window size of 100 kb by step size of 20 kb and threshold of ld of 0.6 (r2)). For reference LD score, tables were calculated using the LDSC software[61] using the 1000 Genomes African Dataset[39] and genetic map (build hg19) as described in the LD Score Estimation Tutorial (https://github.com/bulik/ldsc/wiki/LD-Score-Estimation-Tutorial). We reported non-infinitesimal mixed model association test $P$-values for Stage 1 GWAS. All association tests were based on impute2 dosage. The FUMA online platform[38] was used for partitioning associations into locus, lead SNPs and independent SNPs based on the 1000 Genomes African dataset.

**Stage 2: Meta analysis.** To evaluate the robustness of associations detected in joint analysis of the AWI-Gen dataset, we performed a meta-analysis of summary statistics from the South, East and West African cohort specific analyses using METASOFT (v2.0.1)[62] implemented in https://github.com/h3abionet/h3agwas/ tree/master/meta/meta-assoc.nf. We also performed a meta-analysis of the summary statistics from AWI-Gen data and 4 cohorts AADM, UGR, DDS, DDC included in Gurdasani et al. 2019. We primarily used the Han and Eskin's Random Effects model (RE2), as it has been suggested to best address the heterogeneity in a cohort like ours, but also recorded $P$-values derived from the Fixed Effect (FE) Binary Effects model (BE) for comparison. We restricted the analysis to only those SNPs that were included in the final AWI-Gen dataset and in at least three of the four other cohorts. The FUMA online platform[38], as mentioned above, was used for classifying the associations.

**Assessment of transferability of signals.** Genomic regions associated with each of the four lipid traits in the GLGC and PAGE consortium studies were identified from the respective summary statistics. For each trait, genome-wide significant SNPs in the discovery set, if also present in the AWI-Gen study, were selected. Individual hits, if separated by less than 50 Kb, were merged into a single locus. Using this approach, for the GLGC study we defined 138, 181, 174, 135 independent regions or loci for LDL-C, HDL-C, TC and TG, respectively. Similarly, for the PAGE study 21, 26, 32 and 17 loci were detected for LDL-C, HDL-C, TC and TG, respectively. The loci from the GLGC study were then partitioned based on lowest $P$-value into Very Strong ($P$-value < $10^{-100}$), Strong ($10^{-20} > P$-value > $10^{-100}$) and Moderate ($5 \times 10^{-8} > P$-value > $10^{-20}$) signals. For each locus we recorded the lowest $P$-value that was observed for any of the significant SNPs in the discovery set in each of the 6 replication sets (Stage-2 GWAS, Stage-1 GWAS, UGR, AADM, DDS and DDC). If a locus was represented by a SNP with a $P$-value < $5 \times 10^{-4}$ it was considered as a replication. We also assessed replication at the nominal $P$-value cut-off of 0.05 as well as a genome-wide significant threshold. For PAGE study based signals only two categories – Strong ($P$-value < $10^{-20}$) and Moderate were considered ($5 \times 10^{-8} > P$-value > $10^{-20}$).

**Assessment of heterogeneity among African cohorts.** Minor allele frequencies and effect sizes (along with standard errors) for the four lipid traits in the AADM, UGR, DDS and DDC studies were obtained from the Gurdasani et al.[7] summary statistics file (downloaded from the GWAS catalog). Minor allele frequencies for the AWI-Gen East African, AWI-Gen West African and AWI-Gen Southern African populations were estimated using PLINK[60] and effect sizes were obtained from geographic-region-specific GWAS conducted in these populations.

**Functional analyses.** The FUMA online platform[38] was used to annotate, prioritize, visualize and interpret the GWAS results. This included extensive functional annotation of all SNPs matching their chromosome base-pair position, and reference and alternate alleles to databases containing known functional annotations. Genes implicated by the mapping of association signals were further investigated using the GENE2FUNC procedure in FUMA, which provides

hypergeometric tests of enrichment of the list of mapped genes in 53 GTEx tissue-specific gene expression sets, 7,246 MSigDB gene sets and chromatin states. Finally, the Multi-marker analysis of genomic annotation (MAGMA, v1.6) analysis, implemented in FUMA, was performed using summary statistics of our association results as input. These gene-based analyses enabled summarization of SNP associations at the gene level and association of the set of genes to biological pathways.

**Fine-mapping analysis.** To perform fine-mapping, we used three software packages included in the h3agwas pipeline (https://github.com/h3abionet/h3agwas/ tree/master/finemapping) that employ Bayesian calculation of posterior probability and/or annotation information to identify potential causal variants. For each region around the Stage 1 GWAS signals, using FINEMAP[44], we performed fine-mapping with stepwise conditioning (--cond) and shotgun stochastic search (--sss), providing us with Posterior Inclusion Probabilities (PIP). The PIP for the $l$th SNP is the posterior probability that this SNP is causal and the Bayes factor quantifies the evidence that the $l$th SNP is causal with $\log_{10}$ Bayes factors greater than 2 reporting considerable evidence (all SNPs greater than 2 with a significant $P$-value are included in the credible set). PAINTOR[45] empirically estimated the contribution of each functional annotation to the trait of interest directly from summary association statistics while allowing for multiple causal variants at any risk locus. Additionally, we used CAVIAR-BF[46] for fine-mapping using marginal test statistics in the Bayesian framework. In addition, COJO-GCTA[63] was used to perform a stepwise model selection procedure to select independently associated SNPs (--cojo-slct); and we also performed association analysis of the included SNPs conditional on the given list of SNPs (--cojo-cond).

**Prediction and comparison of polygenic risk score (PRS) models.** The clumping and thresholding (C + T) approach in PRSice2[64] was used for developing PRS. Clumping distance of 250 kb, $r^2$ of 0.1, the optimal $P$-value thresholds for computing the polygenic risk scores for the lipid traits are indicated in the Supplementary Data 8. Lipid-trait GWAS summary statistics from Gurdasani et al.[7], GLGC[15] and PAGE[36] studies were used as the base (discovery) and the AWI-Gen study participants were used as the target dataset. The prediction models were corrected for age, sex and residual population structure using principal components. The R2 (variance) was used to determine the best predictive PRS. The polygenic risk scores were first tested on a third of the AWI-Gen participants ($n = 3500$) and then the best predictive polygenic risk scores were validated in the remaining two-thirds of the AWI-Gen dataset ($n = 7103$). Decile plots were used to evaluate the risk stratification of PRS, by comparing the mean difference of the lipid traits of the upper deciles from the lowest decile.

**Reporting summary.** Further information on research design is available in the Nature Research Reporting Summary linked to this article.

## Data availability
The full dataset generated in this study is in the EGA [https://ega-archive.org/] database under the study accession code EGAS00001002482. This includes the phenotype dataset EGAD00001006425 and the genotype dataset EGAD00010001996. These datasets are available subject to controlled access through the Data and Biospecimen Access Committee of the H3Africa Consortium. The processed data generated in this study are provided in Supplementary Information and Supplementary Data. Summary statistics reported in the paper are accessible on GWAS Catalog (https://www.ebi.ac.uk/gwas/) at the accession numbers: GCST90101741, GCST90101742, GCST90101743, GCST9010174, GCST90101745, GCST90101746, GCST90101747, GCST90101748. All data that support the findings of this study are available from the corresponding authors on request. Publicly available datasets included in the study are the following: 1000 Genomes Project Phase 3 (ftp://ftp.1000genomes.ebi.ac.uk/vol1/ftp), UGR meta-analysis summary statistics, GLGC summary Statistics, PAGE consortium summary statistics available at GWAS Catalog (https://www.ebi.ac.uk/gwas/) and gnomAD (https:// gnomad.broadinstitute.org/).

## Code availability
The H3A-Africa GWAS pipeline was employed for QC, association testing, meta-analysis and fine-mapping are available at (https://github.com/h3abionet/h3agwas) (see Methods).

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

## Acknowledgements

The authors acknowledge the AWI-Gen field workers, phlebotomists, laboratory scientists, administrators, data personnel and all other staff who contributed to the data and sample collections, processing, storage, and shipping, and the participants without whom this work would not have been possible. Investigators responsible for the conception and design of the original AWI-Gen study include the following: Michèle Ramsay (PI, Wits), Osman Sankoh (co-PI, INDEPTH), Alisha Wade, Stephen Tollman and Kathleen Kahn (Agincourt), Marianne Alberts (Dikgale), Catherine Kyobutungi (Nairobi), Halidou Tinto (Nanoro), Abraham Oduro (Navrongo), Shane A. Norris (Soweto), and Scott Hazelhurst, Nigel Crowther, Himla Soodyall and Zane Lombard (Wits). The authors would like to acknowledge each of the following investigators for the significant contributions made to this research: Agincourt: Ryan Wagner and Sulaimon Afolabi; Dikgale: Ian Cook and Sam Ntuli; Nairobi: Stella Muthuri; Nanoro: Toussaint Rouamba and Moussa Lingani; Navrongo: Godfred Agongo, Lucas Amenga-Etego and Engelbert A. Nonterah; Soweto: Nomses Baloyi, Juliana Kagura, Richard Munthali and Yusuf Guman; University of the Witwatersrand/National Health Laboratory Service: Venesa Pillay. We thank Christopher Mathew for his critical reading and comments on the draft manuscript. The AWI-Gen Collaborative Centre is funded by the National Human Genome Research Institute (NHGRI), Office of the Director (OD), Eunice Kennedy Shriver National Institute Of Child Health & Human Development (NICHD), the National Institute of Environmental Health Sciences (NIEHS), the Office of AIDS Research (OAR) and the National Institute of Diabetes and Digestive and Kidney Diseases (NIDDK), of the National Institutes of Health under award number U54HG006938 and its supplements, as part of the H3Africa Consortium. Additional funding was granted by the Department of Science and Technology (now Department of Science and Innovation), South Africa, award number DST/CON 0056/2014. T.C. is an international training fellow supported by the Wellcome Trust grant (214205/Z/18/Z).

## Author contributions

Study Design: M.R., S.H. and A.C.; Data preparation and initial processing: S.H.; D.S.; Data Analysis: A.C., J.-T.B., T.C., D.S., P.R.B. and S.H. Other contributions: F.X.G.-O., S.T., E.M., S.M.-M., I.K., G.A., H.S., H.T., G.A.g., E.A.N., S.N. and L.M. directed the field work and sample collection. S.G. and C.J.W. performed look-ups in the GLGC African ancestry and trans-ethnic GWAS. Writing: A.C. drafted the manuscript based on inputs from J.T., D.S., T.C., P.R.B., N.C., S.H. and M.R., and additional editing from co-authors. All authors critically evaluated and approved the manuscript.

## Competing interests

The authors declare no competing interests.

## Additional information

## AWI-Gen study

Ananyo Choudhury[1,16✉], Jean-Tristan Brandenburg[1,16], Tinashe Chikowore[1,2], Dhriti Sengupta[1], Palwende Romuald Boua[1,3], Nigel J. Crowther[4], Godfred Agongo[5,6], Gershim Asiki[7], F. Xavier Gómez-Olivé[8], Isaac Kisiangani[7], Eric Maimela[9], Matshane Masemola-Maphutha[10], Lisa K. Micklesfield[2], Engelbert A. Nonterah[5], Shane A. Norris[2], Hermann Sorgho[3], Halidou Tinto[3], Stephen Tollman[8], Scott Hazelhurst[1,14] & Michèle Ramsay[1,15✉]

## H3Africa Consortium

Ananyo Choudhury[1,16✉], Jean-Tristan Brandenburg[1,16], Tinashe Chikowore[1,2], Dhriti Sengupta[1], Palwende Romuald Boua[1,3], Nigel J. Crowther[4], Godfred Agongo[5,6], Gershim Asiki[7], F. Xavier Gómez-Olivé[8], Isaac Kisiangani[7], Lisa K. Micklesfield[2], Engelbert A. Nonterah[5], Shane A. Norris[2], Hermann Sorgho[3], Halidou Tinto[3], Stephen Tollman[8], Scott Hazelhurst[1,14] & Michèle Ramsay[1,15✉]

