## [Peer Review File · Nature Communications]

Meta-analysis of sub-Saharan African studies provides insights into genetic architecture of lipid traitsReviewers' Comments:

Reviewer #1:

Remarks to the Author:

Comments to the author:

This study performed meta-analysis of lipid traits in 25000 Africans, including ~11000 individuals newly genotyped by this study and ~14000 individuals from published studies. Main findings of this paper include (1) two novel association signals specific to African populations; (2) demonstrating the advantages of including diverse African populations to facilitate fine-mapping and to enhance transferability of PRS. This paper contributes important human genetics data in the currently underrepresented African populations. The manuscript is well written and the conclusions are well supported by the data.

My major concern of this paper is that its novelty might be significantly compromised because the GLGC paper has been accepted by Nature. The key results (and the analyses) of showing the advantage in fine-mapping and transferability of PRS by including diverse populations are very similar in the two papers, except that GLGC has a much larger sample size by including other ancestral groups. The data reported in this paper are included in the GLGC paper. The authors of this paper are also in the author list of the GLGC paper. Comparing the two papers, this paper has a unique focus on the African-specific association signals. But the signals of the two novel loci are relatively weak (p values are very close to the threshold of $5e-8$) and have no other data to support the functional role of these two novel loci.

Specific comments:

- 1.The section "Fine-mapping of lipid trait loci" only provides two examples of PCSK9 and LDLR. How about the other lipid loci? It will be good to provide an overall summary of all loci (and list full results in the Supplements) to illustrate the benefits of including African data in fine-mapping.
- 2.Please use a consistent gene name for FHIP1A/FAM160A1. Switching different names in Figure 2 and the text can create confusion.
- 3.It will be good to use the same color scheme in panels a and b of Figure 5.
- 4.A typo on line 374: "an continental ..." should be "a continental...".

Reviewer #2:

Remarks to the Author:

In the present manuscript, Choudhury et al. perform Meta-analysis using the AWI-Gen cohort and existing summary statistics from published large trans-ethnic GWAS on lipid traits in African ancestry population.

This is an important study contributing toward addressing the problem of health disparities across diverse populations for cardiometabolic diseases. The methodological approach to establish association signals in Meta-analysis using published GWAS data and evaluating them using Polygenic risk scores approach is sound.

Minor comments:

- The major analytical method used is Meta-analysis, It would be useful for the readers to provide additional details about the Meta-analysis method. For example, Han and Eskin's Random Effects

model (RE2) was used. It would be useful to state that this method accounts for heterogeneity highlighted in lines 425-430. Moreover, the github link for Metasoft implementation (<http://github.com/h3abionet/h3agwas/meta/meta.nf>) is not working.

- The paper lacks emphasis on clinical implications/benefits of this study. Clinical implication of a specific signal is discussed (lines 480-484) but the Discussion should also focus on potential clinical implications in general. Which interventions are available and how would patients benefit from this study especially in terms of cardiometabolic diseases?
- For PRS calculations, it will be useful for the readers if AUC (AUROC) plots highlighting the lipid traits are also included.
- It is not clear what unit is used for P-value, SE and Alt_AF columns in S Table 8. Looks like “,” should be replaced by “.”; Also the P-value column in Trans-ancestry analysis (METAL) section need to be fixed. Similarly for supplementary tables 6 and 7.

Reviewer #3:

Remarks to the Author:

Thank you for the opportunity to review the manuscript, “Meta-analysis of ~25,000 continental Africans provides insights into the genetic architecture of lipid traits,” by Choudhury and colleagues. This analysis is the largest GWAS to date of lipids traits across multiple African populations. The authors use the H3Africa SNP genotyping array with imputation to the African Reference Panel to perform GWAS for fasting serum total cholesterol, LDL cholesterol, HDL cholesterol, and triglycerides among >10,000 participants of the AWI-Gen cohort. In their primary discovery analysis, they identify a novel genome-wide significant (GWS) SNP for LDL-C. Although this SNP drops just below the GWS threshold in the meta-analysis with outside data, additional linked SNPs in the same region rise to GWS. Overall, I believe that the authors provide compelling evidence for discovery of a novel LDL-C associated locus. They also provide a useful comparison of joint analysis using a linear mixed model versus the traditional stratified analysis followed by meta-analysis. Beyond GWAS, they provide additional findings that are of interest to the field. The authors demonstrate the power of African populations to improve fine mapping of known loci, and they provide evidence that previous observations of low transferability of lipid GWAS is likely driven in large part by lack of power, further supporting the need for larger GWAS of African populations. The manuscript is generally quite clear and well written. However, I think there are a few key issues that should be addressed prior to publication, as outlined below.

MAJOR

1. The comparison of joint analysis versus stratified analysis with meta-analysis is interesting and informative. Can the authors please provide additional details of this comparison either in the main text or the supplement? Specifically, how many genome-wide significant (GWS) SNPs were identified by each approach, and how many SNPs were GWS by both approaches. Further, how many independent loci were identified by each approach, and how many were identified by both? Put another way, were any loci uniquely identified by one approach or the other? Although Table 1 provides some of this information, it was unclear to me if all lead SNPs from both approaches are listed in that table.
2. The authors use clumping and thresholding to generate polygenic scores for the four lipid traits. They compare scores created using summary statistics from GLGC, PAGE, and Gurdasani et al. 2019. They describe a tuning phase to identify the optimal p-value threshold. However, there are a few

important details of their approach that are unclear and may require improvement. First, what is the LD reference panel used to determine r^2 values for each set of summary statistics? Second, was the tuning process to determine optimal p-value threshold performed in a separate/independent cohort from that used for validation analyses (eg Figure 5)? I believe this type of analysis is most compelling and most robust when the tuning cohort and validation cohorts are separate. If this is not the case, I would request that the authors divide the AWI-Gen subjects into a tuning set and a validation set. To my knowledge, there is no clear best practice for how to divide the cohort, so I leave that to the author's discretion, but one example would be to randomly select $\sim 1/3$ or $\sim 1/4$ of the cohort to use for tuning and then use the rest for validation.

3. Relating to point 2 above, please report the optimal p-value threshold for each set of summary statistics and for each lipid trait. Since the primary focus of this manuscript is not generation of polygenic scores, I think it is reasonable to use the simple approach of p-value optimization. Though, it should be noted that only a single set of clumping parameters were considered.

MINOR

1. The introduction specifies that the cohort age is >40 years, but Supplemental Table 1 indicates participants as young as 29 years. Is this a typo in the introduction?

2. The mean LDL in the West African cohorts is strikingly low. I am unfamiliar with typical LDL values in different African cohorts, so I would be curious to know if this is a previously observed phenomenon or if there is a plausible explanation for the difference.

3. The authors note that their GWAS includes ~ 14 M SNPs, while the Gurdasani GWAS included ~ 22 M SNPs. Can the authors add 1-2 sentences explaining the difference (eg different imputation panel?).

4. Table 1 has inconsistent notation for specifying the decimal place (comma versus period).

5. For Figure 5, is there a strong reason to place total cholesterol in the supplement, with LDL, HDL, and triglycerides in the main? I defer to the authors and editors, but I suspect many readers would appreciate seeing the results for all four traits together.

6. In the main text or in the legend of Figure 5, it would be helpful to report the GWAS sample size for each set of summary statistics.

7. In the Discussion, the authors describe a look up of their SNPs in a very large GLGC GWAS. Are these unpublished results? If so, this fact should be specified. Otherwise, please provide a reference.

8. Supplemental Table 8 appears to list incorrect p values for the METAL analysis (perhaps missing leading 0?).

REVIEWER COMMENTS

Reviewer #1 (Remarks to the Author)

Comments to the author:

This study performed meta-analysis of lipid traits in 25000 Africans, including ~11000 individuals newly genotyped by this study and ~14000 individuals from published studies. Main findings of this paper include (1) two novel association signals specific to African populations; (2) demonstrating the advantages of including diverse African populations to facilitate fine-mapping and to enhance transferability of PRS. This paper contributes important human genetics data in the currently underrepresented African populations. The manuscript is well written and the conclusions are well supported by the data.

1. My major concern of this paper is that its novelty might be significantly compromised because the GLGC paper has been accepted by Nature. The key results (and the analyses) of showing the advantage in fine-mapping and transferability of PRS by including diverse populations are very similar in the two papers, except that GLGC has a much larger sample size by including other ancestral groups. The data reported in this paper are included in the GLGC paper. The authors of this paper are also in the author list of the GLGC paper. Comparing the two papers, this paper has a unique focus on the African-specific association signals. But the signals of the two novel loci are relatively weak (p values are very close to the threshold of $5e-8$) and have no other data to support the functional role of these two novel loci.

Response: We understand the reviewer's concern and would like to clarify that AWI-Gen is not a constituent cohort of the GLGC Nature paper (Grahams et al. 2021) and their meta-analysis does not include our summary stats. During the review of the GLGC paper, when the authors were requested for additional data to support their PRS analysis, they approached AWI-Gen for a lookup of the two PRSs developed by them. So, for the GLGC paper we have only performed lookup for their PRSs and our data features in only one of their figures. Therefore, our data is not included in the GWAS, fine-mapping, PRS development or any of the other analyses.

Similarly, we have not incorporated the Graham et al. 2021 datasets in our GWAS or meta-analysis and have only requested for a look-up of the 3 potentially novel signals identified in our study, in the GLGC summary stats. The reciprocal lookups form the basis of our authorship in the GLGC paper and the authorship of two GLGC authors in the current paper.

We would like to highlight that the key differences between the two papers, which makes our study novel, is that the GLGC study has very limited continental African representation (most of the African ancestry participants were African Americans) whereas our paper is based completely on continental African cohorts and does not include any African American datasets. Moreover, the focus of the PRS analysis was also different. While the GLGC study aimed to develop PRSs, our goal was to assess the performance of existing PRSs in continental African populations. Furthermore, in the current manuscript we have not tested for the PRSs developed in the Grahams et al. 2021 study, to avoid any redundancy.

Specific comments:

1. The section “Fine-mapping of lipid trait loci” only provides two examples of PCSK9 and LDLR. How about the other lipid loci? It will be good to provide an overall summary of all loci (and list full results in the Supplements) to illustrate the benefits of including African data in fine-mapping.

Response: To address this concern we have included three new Supplementary Figures (S Figures 8-11 in the current manuscript) that contain additional Locuszoom plots for our results. Also, we have included Locuszoom plots for the same regions based on the Prins et al. 2017 dataset. The comparison, as suggested, shows better fine-mapping of several lipid loci with African data. We have amended the text in the fine-mapping section to include these examples of fine mapping as follows –

“Although the top SNPs from a locus often differed between the two datasets, several other well-known lipid loci such as *LPAL2*, *LIPC*, *CELSR2* and *APOC1* showed a narrower peak in the AWI-Gen GWAS in comparison to the European GWAS study (S Figures 8-11).”

2. Please use a consistent gene name for FHIP1A/FAM160A1. Switching different names in Figure 2 and the text can create confusion.

Response: We thank the reviewer for directing us to the discrepancy, we now refer to this gene as *FHIP1A* throughout the paper and the figures have also been amended accordingly.

3. It will be good to use the same color scheme in panels a and b of Figure 5.

Response: We thank the reviewer for this observation, Figure 5 a-h has been amended to include the same colour scheme and we have now added the colour code to the figure legend.

4. A typo on line 374: “an continental ...” should be “a continental...”.

Response: We thank the reviewer for the careful reading of the manuscript. We have amended the text.

Reviewer #2 (Remarks to the Author):

In the present manuscript, Choudhury et al. perform Meta-analysis using the AWI-Gen cohort and existing summary statistics from published large trans-ethnic GWAS on lipid traits in African ancestry population.

This is an important study contributing toward addressing the problem of health disparities across diverse populations for cardiometabolic diseases. The methodological approach to establish association signals in Meta-analysis using published GWAS data and evaluating them using Polygenic risk scores approach is sound.

Minor comments:

1. The major analytical method used is Meta-analysis, It would be useful for the readers to provide additional details about the Meta-analysis method. For example, Han and Eskin's Random Effects model (RE2) was used. It would be useful to state that this method accounts for heterogeneity highlighted in lines 425-430. Moreover, the github link for Metasoft implementation (<http://github.com/h3abionet/h3agwas/meta/meta.nf>) is not working.

Response: We agree with the reviewer's recommendations for additional details on the Meta-soft model. We have modified (shown in blue) the following line in the methods section to highlight this:

"We primarily used the Han and Eskin's Random Effects model (RE2), **as it has been suggested to best address the heterogeneity in a cohort like ours**, but also recorded P -values derived from the Fixed Effect (FE) Binary Effects model (BE) for comparison."

We also sincerely appreciate the detection of the broken link. We have updated the link in the manuscript (<https://github.com/h3abionet/h3agwas/tree/master/meta/meta-assoc.nf>).

2. The paper lacks emphasis on clinical implications/benefits of this study. Clinical implication of a specific signal is discussed (lines 480-484) but the Discussion should also focus on potential clinical implications in general. Which interventions are available and how would patients benefit from this study especially in terms of cardiometabolic diseases?

Response: We accept the reviewers' criticism on the lack of discussion on the clinical implications. Since this paper describes a GWAS for lipid traits it has limited application in terms of direct clinical implications. This is primarily because the predictive value of GWAS and Polygenic Risk Score (PRS) models for lipid levels remains low, as it does for most complex cardio-metabolic traits, especially in non-European populations. Furthermore, we need to be cautious about clinical implications/benefits, given that this is a modest GWAS sample (although the largest in Africa) and that there is much regional variation in Africa. We have touched upon some of these in the 3rd last paragraph of discussion. At this time, unfortunately, individual participants (or future patients) will not be able to benefit from the results at a clinical level. However, we do expect that with more such studies we would gradually be able to generate models that could enable risk assessment and patient prioritization for treatment in these populations.

3 For PRS calculations, it will be useful for the readers if AUC (AUROC) plots highlighting the lipid traits are also included.

Response: We agree with the reviewer's point that AUC plots are indeed helpful in showing the performance, in terms of specificity and sensitivity, of the PRSs for a trait. However, due to lack of previous large-scale data from African populations, it is quite challenging to decide on cut-offs that are appropriate for defining dyslipidaemia (as a binary trait) in these populations. Therefore, we have limited all our analyses to quantitative lipid traits and used the variance (R^2) as a proxy for predictivity of the polygenic risk score and not AUC that applies to binary traits.

4. It is not clear what unit is used for P-value, SE and Alt_AF columns in S Table 8. Looks like “,” should be replaced by “.”; Also the P-value column in Trans-ancestry analysis (METAL) section need to be fixed. Similarly for supplementary tables 6 and 7.

Response: We sincerely thank the reviewer for pointing out the error in S Table 8 (in the previous version). There was a shift in the location of the P-value and the sample size columns. Also, the “.” got replaced by commas. We have amended all three supplementary tables (6,7 and 8 in the previous version) accordingly.

Reviewer #3 (Remarks to the Author):

Thank you for the opportunity to review the manuscript, “Meta-analysis of ~25,000 continental Africans provides insights into the genetic architecture of lipid traits,” by Choudhury and colleagues. This analysis is the largest GWAS to date of lipids traits across multiple African populations. The authors use the H3Africa SNP genotyping array with imputation to the African Reference Panel to perform GWAS for fasting serum total cholesterol, LDL cholesterol, HDL cholesterol, and triglycerides among >10,000 participants of the AWI-Gen cohort. In their primary discovery analysis, they identify a novel genome-wide significant (GWS) SNP for LDL-C. Although this SNP drops just below the GWS threshold in the meta-analysis with outside data, additional linked SNPs in the same region rise to GWS. Overall, I believe that the authors provide compelling evidence for discovery of a novel LDL-C associated locus. They also provide a useful comparison of joint analysis using a linear mixed model versus the traditional stratified analysis followed by meta-analysis. Beyond GWAS, they provide additional findings that are of interest to the field. The authors demonstrate the power of African populations to improve fine mapping of known loci, and they provide evidence that previous observations of low transferability of lipid GWAS is likely driven in large part by lack of power, further supporting the need for larger GWAS of African populations. The manuscript is generally quite clear and well written. However, I think there are a few key issues that should be addressed prior to publication, as outlined below.

MAJOR

1. The comparison of joint analysis versus stratified analysis with meta-analysis is interesting and informative. Can the authors please provide additional details of this comparison either in the main text or the supplement? Specifically, how many genome-wide significant (GWS) SNPs were identified by each approach, and how many SNPs were GWS by both approaches. Further, how many independent loci were identified by each approach, and how many were identified by both? Put another way, were any loci uniquely identified by one approach or the other? Although Table 1 provides some of this information, it was unclear to me if all lead SNPs from both approaches are listed in that table.

Response: We agree with the reviewer that more information on the comparison of joint versus meta-analysis performed on the AWI-Gen dataset is useful. Based on the reviewer’s

suggestion we have added a new supplementary table summarizing the results from the two approaches (new Supplementary Table 2). As evident from the table, the two approaches identify exactly the same signals at the locus level. Even at the SNP level there is a high-overlap between the independent significant SNPs and lead SNPs.

2. The authors use clumping and thresholding to generate polygenic scores for the four lipid traits. They compare scores created using summary statistics from GLGC, PAGE, and Gurdasani et al. 2019. They describe a tuning phase to identify the optimal p-value threshold. However, there are a few important details of their approach that are unclear and may require improvement. First, what is the LD reference panel used to determine r^2 values for each set of summary statistics? Second, was the tuning process to determine optimal p-value threshold performed in a separate/independent cohort from that used for validation analyses (eg Figure 5)? I believe this type of analysis is most compelling and most robust when the tuning cohort and validation cohorts are separate. If this is not the case, I would request that the authors divide the AWI-Gen subjects into a tuning set and a validation set. To my knowledge, there is no clear best practice for how to divide the cohort, so I leave that to the author's discretion, but one example would be to randomly select $\sim 1/3$ or $\sim 1/4$ of the cohort to use for tuning and then use the rest for validation.

Response: We appreciated these helpful comments. We used the AWI-Gen genotype data as the LD reference for determining the optimal r^2 for clumping. Based on the reviewer's advice we have done $1/3^{\text{rd}}$ (tuning set, $N=3500$) and $2/3^{\text{rd}}$ (validation set, $N=7103$) split and performed a two-step PRS analysis. The figure 5 has been updated, S Table 8 has been added to provide detailed stats. Also, the main text and methods have been updated.

3. Relating to point 2 above, please report the optimal p-value threshold for each set of summary statistics and for each lipid trait. Since the primary focus of this manuscript is not generation of polygenic scores, I think it is reasonable to use the simple approach of p-value optimization. Though, it should be noted that only a single set of clumping parameters were considered.

Response: The optimal P-value thresholds for the discovery data sets for the related traits have now been included (see S Table 8). After preliminary analysis we realised that optimal clumping parameters were r^2 of 0.1 and a clumping distance of 250 kb and these were used for the computation of the polygenic risk scores.

MINOR

1. The introduction specifies that the cohort age is >40 years, but Supplemental Table 1 indicates participants as young as 29 years. Is this a typo in the introduction?

Response: We sincerely appreciate this observation. Although our cohort is predominantly 40+, we do have a handful of participants below 40. We have amended the line in the introduction accordingly.

2. The mean LDL in the West African cohorts is strikingly low. I am unfamiliar with typical

LDL values in different African cohorts, so I would be curious to know if this is a previously observed phenomenon or if there is a plausible explanation for the difference.

Response: The reviewer is correct in observing that LDL-C levels in the two West African sites are much lower compared to other four sites. In addition to genetics, a major factor contributing to this difference might be lifestyle and diet. The two West African sites are rural in contrast to the urban or semi-urban nature of the other four sites, which could also contribute to this difference. Also, we have observed similarly lower prevalence for traits such as hypertension, obesity and diabetes in the two west African sites compared to the other four sites, suggesting that this trend includes several cardiometabolic traits (Gomez Olive et al. 2017, Nonterah et al. 2019, Ramsay et al. 2018, Pisa et al. 2018). Lower lipid levels in West Africans in comparison to other populations including Admixed Africans has also been recorded (Bentley and Rotimi 2017).

A study to explore the difference in levels of lipid-markers and the identification of lifestyle, demographics and other possible factors associated with them is being performed in an independent study by other members of the AWI-Gen group.

3. The authors note that their GWAS includes ~14 M SNPs, while the Gurdasani GWAS included ~22 M SNPs. Can the authors add 1-2 sentences explaining the difference (eg different imputation panel?).

Response: We thank the reviewer for highlighting this difference in numbers. We have added the following line in Page 5 to address this:

“The difference, despite the use of the same imputation panel, was largely due to a more stringent post imputation MAF cut-off (MAF>0.01 in AWI-Gen) compared to that used in Gurdasani et al. 2019⁷ (MAF>0.005).”

4. Table 1 has inconsistent notation for specifying the decimal place (comma versus period).
[11]

Response: We thank the reviewer for detecting the error. We have amended the table so as all the decimal places are indicated by periods.

5. For Figure 5, is there a strong reason to place total cholesterol in the supplement, with LDL, HDL, and triglycerides in the main? I defer to the authors and editors, but I suspect many readers would appreciate seeing the results for all four traits together.

Response: Thanks for the observations. Indeed, there was no reason for placing the TC in the S Figure and the other 3 traits in the main figure, beyond ensuring better visibility. Based on the reviewer's recommendation we have added all the four traits in Figure 5.

6. In the main text or in the legend of Figure 5, it would be helpful to report the GWAS sample size for each set of summary statistics.

Response: We agree that displaying sample sizes in the figure legend is useful. Therefore, based on the reviewer's recommendation we have added the sample sizes in the legend.

7. In the Discussion, the authors describe a look up of their SNPs in a very large GLGC GWAS. Are these unpublished results? If so, this fact should be specified. Otherwise, please provide a reference.

Response: Thank you for this observation. Since submitting the manuscript, the GLGC study has been published (Grahams et al. 2021) and we have added the reference in the current version of the manuscript.

8. Supplemental Table 8 appears to list incorrect p values for the METAL analysis (perhaps missing leading 0?)

Response: We thank the reviewer for pointing this out. There was a column shift, due to which the P-value column got replaced by sample size. We have amended this Supplementary Table (now S Table 10) accordingly.

Reviewers' Comments:

Reviewer #1:

Remarks to the Author:
Happy to authors.

Reviewer #2:

Remarks to the Author:
Thank you for addressing the comments.

Reviewer #3:

Remarks to the Author:

I appreciate the opportunity to review this revised submission of the manuscript, "Meta-analysis of ~25,000 continental Africans provides insights into the genetic architecture of lipid traits," by Choudhury and colleagues. In their revision, the authors have thoroughly addressed my prior comments, and I also found their comments in response to the reviewers to be very informative. Overall, this is an important paper with novel insights and sound methods. I recommend acceptance of the current revision.

REVIEWER COMMENTS (Round 1)

Reviewer #1 (Remarks to the Author)

Comments to the author:

This study performed meta-analysis of lipid traits in 25000 Africans, including ~11000 individuals newly genotyped by this study and ~14000 individuals from published studies. Main findings of this paper include (1) two novel association signals specific to African populations; (2) demonstrating the advantages of including diverse African populations to facilitate fine-mapping and to enhance transferability of PRS. This paper contributes important human genetics data in the currently underrepresented African populations. The manuscript is well written and the conclusions are well supported by the data.

1. My major concern of this paper is that its novelty might be significantly compromised because the GLGC paper has been accepted by Nature. The key results (and the analyses) of showing the advantage in fine-mapping and transferability of PRS by including diverse populations are very similar in the two papers, except that GLGC has a much larger sample size by including other ancestral groups. The data reported in this paper are included in the GLGC paper. The authors of this paper are also in the author list of the GLGC paper. Comparing the two papers, this paper has a unique focus on the African-specific association signals. But the signals of the two novel loci are relatively weak (p values are very close to the threshold of $5e-8$) and have no other data to support the functional role of these two novel loci.

Response: We understand the reviewer's concern and would like to clarify that AWI-Gen is not a constituent cohort of the GLGC Nature paper (Grahams et al. 2021) and their meta-analysis does not include our summary stats. During the review of the GLGC paper, when the authors were requested for additional data to support their PRS analysis, they approached AWI-Gen for a lookup of the two PRSs developed by them. So, for the GLGC paper we have only performed lookup for their PRSs and our data features in only one of their figures. Therefore, our data is not included in the GWAS, fine-mapping, PRS development or any of the other analyses.

Similarly, we have not incorporated the Graham et al. 2021 datasets in our GWAS or meta-analysis and have only requested for a look-up of the 3 potentially novel signals identified in our study, in the GLGC summary stats. The reciprocal lookups form the basis of our authorship in the GLGC paper and the authorship of two GLGC authors in the current paper.

We would like to highlight that the key differences between the two papers, which makes our study novel, is that the GLGC study has very limited continental African representation (most of the African ancestry participants were African Americans) whereas our paper is based completely on continental African cohorts and does not include any African American datasets. Moreover, the focus of the PRS analysis was also different. While the GLGC study aimed to develop PRSs, our goal was to assess the performance of existing PRSs in continental African populations. Furthermore, in the current manuscript we have not tested for the PRSs developed in the Grahams et al. 2021 study, to avoid any redundancy.

Specific comments:

1. The section “Fine-mapping of lipid trait loci” only provides two examples of PCSK9 and LDLR. How about the other lipid loci? It will be good to provide an overall summary of all loci (and list full results in the Supplements) to illustrate the benefits of including African data in fine-mapping.

Response: To address this concern we have included three new Supplementary Figures (**S Figures 8-11** in the current manuscript) that contain additional Locuszoom plots for our results. Also, we have included Locuszoom plots for the same regions based on the Prins et al. 2017 dataset. The comparison, as suggested, shows better fine-mapping of several lipid loci with African data. We have amended the text in the fine-mapping section to include these examples of fine mapping as follows –

“Although the top SNPs from a locus often differed between the two datasets, several other well-known lipid loci such as *LPAL2*, *LIPC*, *CELSR2* and *APOC1* showed a narrower peak in the AWI-Gen GWAS in comparison to the European GWAS study (**S Figures 8-11**).”

2. Please use a consistent gene name for FHIP1A/FAM160A1. Switching different names in Figure 2 and the text can create confusion.

Response: We thank the reviewer for directing us to the discrepancy, we now refer to this gene as *FHIP1A* throughout the paper and the figures have also been amended accordingly.

3. It will be good to use the same color scheme in panels a and b of Figure 5.

Response: We thank the reviewer for this observation, Figure 5 a-h has been amended to include the same colour scheme and we have now added the colour code to the figure legend.

4. A typo on line 374: “an continental ...” should be “a continental...”.

Response: We thank the reviewer for the careful reading of the manuscript. We have amended the text.

Reviewer #2 (Remarks to the Author):

In the present manuscript, Choudhury et al. perform Meta-analysis using the AWI-Gen cohort and existing summary statistics from published large trans-ethnic GWAS on lipid traits in African ancestry population.

This is an important study contributing toward addressing the problem of health disparities across diverse populations for cardiometabolic diseases. The methodological approach to establish association signals in Meta-analysis using published GWAS data and evaluating them using Polygenic risk scores approach is sound.

Minor comments:

1. The major analytical method used is Meta-analysis, It would be useful for the readers to provide additional details about the Meta-analysis method. For example, Han and Eskin's Random Effects model (RE2) was used. It would be useful to state that this method accounts for heterogeneity highlighted in lines 425-430. Moreover, the github link for Metasoft implementation (<http://github.com/h3abionet/h3agwas/meta/meta.nf>) is not working.

Response: We agree with the reviewer's recommendations for additional details on the Meta-soft model. We have modified (shown in blue) the following line in the methods section to highlight this:

“We primarily used the Han and Eskin's Random Effects model (RE2), **as it has been suggested to best address the heterogeneity in a cohort like ours**, but also recorded *P*-values derived from the Fixed Effect (FE) Binary Effects model (BE) for comparison.”

We also sincerely appreciate the detection of the broken link. We have updated the link in the manuscript (<https://github.com/h3abionet/h3agwas/tree/master/meta/meta-assoc.nf>).

2. The paper lacks emphasis on clinical implications/benefits of this study. Clinical implication of a specific signal is discussed (lines 480-484) but the Discussion should also focus on potential clinical implications in general. Which interventions are available and how would patients benefit from this study especially in terms of cardiometabolic diseases?

Response: We accept the reviewers' criticism on the lack of discussion on the clinical implications. Since this paper describes a GWAS for lipid traits it has limited application in terms of direct clinical implications. This is primarily because the predictive value of GWAS and Polygenic Risk Score (PRS) models for lipid levels remains low, as it does for most complex cardio-metabolic traits, especially in non-European populations. Furthermore, we need to be cautious about clinical implications/benefits, given that this is a modest GWAS sample (although the largest in Africa) and that there is much regional variation in Africa. We have touched upon some of these in the 3rd last paragraph of discussion. At this time, unfortunately, individual participants (or future patients) will not be able to benefit from the results at a clinical level. However, we do expect that with more such studies we would gradually be able to generate models that could enable risk assessment and patient prioritization for treatment in these populations.

3 For PRS calculations, it will be useful for the readers if AUC (AUROC) plots highlighting the lipid traits are also included.

Response: We agree with the reviewer's point that AUC plots are indeed helpful in showing the performance, in terms of specificity and sensitivity, of the PRSs for a trait. However, due to lack of previous large-scale data from African populations, it is quite challenging to decide on cut-offs that are appropriate for defining dyslipidaemia (as a binary trait) in these populations. Therefore, we have limited all our analyses to quantitative lipid traits and used the variance (R^2) as a proxy for predictivity of the polygenic risk score and not AUC that applies to binary traits.

4. It is not clear what unit is used for P-value, SE and Alt_AF columns in S Table 8. Looks like “,” should be replaced by “.”; Also the P-value column in Trans-ancestry analysis (METAL) section need to be fixed. Similarly for supplementary tables 6 and 7.

Response: We sincerely thank the reviewer for pointing out the error in S Table 8 (in the previous version). There was a shift in the location of the P-value and the sample size columns. Also, the “,” got replaced by commas. We have amended all three supplementary tables (6,7 and 8 in the previous version) accordingly.

Reviewer #3 (Remarks to the Author):

Thank you for the opportunity to review the manuscript, “Meta-analysis of ~25,000 continental Africans provides insights into the genetic architecture of lipid traits,” by Choudhury and colleagues. This analysis is the largest GWAS to date of lipids traits across multiple African populations. The authors use the H3Africa SNP genotyping array with imputation to the African Reference Panel to perform GWAS for fasting serum total cholesterol, LDL cholesterol, HDL cholesterol, and triglycerides among >10,000 participants of the AWI-Gen cohort. In their primary discovery analysis, they identify a novel genome-wide significant (GWS) SNP for LDL-C. Although this SNP drops just below the GWS threshold in the meta-analysis with outside data, additional linked SNPs in the same region rise to GWS. Overall, I believe that the authors provide compelling evidence for discovery of a novel LDL-C associated locus. They also provide a useful comparison of joint analysis using a linear mixed model versus the traditional stratified analysis followed by meta-analysis. Beyond GWAS, they provide additional findings that are of interest to the field. The authors demonstrate the power of African populations to improve fine mapping of known loci, and they provide evidence that previous observations of low transferability of lipid GWAS is likely driven in large part by lack of power, further supporting the need for larger GWAS of African populations. The manuscript is generally quite clear and well written. However, I think there are a few key issues that should be addressed prior to publication, as outlined below.

MAJOR

1. The comparison of joint analysis versus stratified analysis with meta-analysis is interesting and informative. Can the authors please provide additional details of this comparison either in the main text or the supplement? Specifically, how many genome-wide significant (GWS) SNPs were identified by each approach, and how many SNPs were GWS by both approaches. Further, how many independent loci were identified by each approach, and how many were identified by both? Put another way, were any loci uniquely identified by one approach or the other? Although Table 1 provides some of this information, it was unclear to me if all lead SNPs from both approaches are listed in that table.

Response: We agree with the reviewer that more information on the comparison of joint versus meta-analysis performed on the AWI-Gen dataset is useful. Based on the reviewer’s suggestion we have added a new supplementary table summarizing the results from the two approaches (new Supplementary Table 2). As evident from the table, the two approaches identify exactly the same signals at the locus level. Even at the SNP level there is a high-overlap between the independent significant SNPs and lead SNPs.

2. The authors use clumping and thresholding to generate polygenic scores for the four lipid traits. They compare scores created using summary statistics from GLGC, PAGE, and Gurdasani et al. 2019. They describe a tuning phase to identify the optimal p-value threshold. However, there are a few important details of their approach that are unclear and may require improvement. First, what is the LD reference panel used to determine r^2 values for each set of summary statistics? Second, was the tuning process to determine optimal p-value threshold performed in a separate/independent cohort from that used for validation analyses (eg Figure 5)? I believe this type of analysis is most compelling and most robust when the tuning cohort and validation cohorts are separate. If this is not the case, I would request that the authors

divide the AWI-Gen subjects into a tuning set and a validation set. To my knowledge, there is no clear best practice for how to divide the cohort, so I leave that to the author's discretion, but one example would be to randomly select ~1/3 or ~1/4 of the cohort to use for tuning and then use the rest for validation.

Response: We appreciated these helpful comments. We used the AWI-Gen genotype data as the LD reference for determining the optimal r^2 for clumping. Based on the reviewer's advice we have done 1/3rd (tuning set, N=3500) and 2/3rd (validation set, N=7103) split and performed a two-step PRS analysis. The figure 5 has been updated, S Table 8 has been added to provide detailed stats. Also, the main text and methods have been updated.

3. Relating to point 2 above, please report the optimal p-value threshold for each set of summary statistics and for each lipid trait. Since the primary focus of this manuscript is not generation of polygenic scores, I think it is reasonable to use the simple approach of p-value optimization. Though, it should be noted that only a single set of clumping parameters were considered.

Response: The optimal P-value thresholds for the discovery data sets for the related traits have now been included (see S Table 8). After preliminary analysis we realised that optimal clumping parameters were r^2 of 0.1 and a clumping distance of 250 kb and these were used for the computation of the polygenic risk scores.

MINOR

1. The introduction specifies that the cohort age is >40 years, but Supplemental Table 1 indicates participants as young as 29 years. Is this a typo in the introduction?

Response: We sincerely appreciate this observation. Although our cohort is predominantly 40+, we do have a handful of participants below 40. We have amended the line in the introduction accordingly.

2. The mean LDL in the West African cohorts is strikingly low. I am unfamiliar with typical LDL values in different African cohorts, so I would be curious to know if this is a previously observed phenomenon or if there is a plausible explanation for the difference.

Response: The reviewer is correct in observing that LDL-C levels in the two West African sites are much lower compared to other four sites. In addition to genetics, a major factor contributing to this difference might be lifestyle and diet. The two West African sites are rural in contrast to the urban or semi-urban nature of the other four sites, which could also contribute to this difference. Also, we have observed similarly lower prevalence for traits such as hypertension, obesity and diabetes in the two west African sites compared to the other four sites, suggesting that this trend includes several cardiometabolic traits (Gomez Olive et al. 2017, Nonterah et al. 2019, Ramsay et al. 2018, Pisa et al. 2018). Lower lipid levels in West Africans in comparison to other populations including Admixed Africans has also been recorded (Bentley and Rotimi 2017).

A study to explore the difference in levels of lipid-markers and the identification of lifestyle, demographics and other possible factors associated with them is being performed in an independent study by other members of the AWI-Gen group.

3. The authors note that their GWAS includes ~14 M SNPs, while the Gurdasani GWAS included ~22 M SNPs. Can the authors add 1-2 sentences explaining the difference (eg different imputation panel?).

Response: We thank the reviewer for highlighting this difference in numbers. We have added the following line in Page 5 to address this:

“The difference, despite the use of the same imputation panel, was largely due to a more stringent post imputation MAF cut-off (MAF>0.01 in AWI-Gen) compared to that used in Gurdasani et al. 2019⁷ (MAF>0.005).”

4. Table 1 has inconsistent notation for specifying the decimal place (comma versus period). [11]

Response: We thank the reviewer for detecting the error. We have amended the table so as all the decimal places are indicated by periods.

5. For Figure 5, is there a strong reason to place total cholesterol in the supplement, with LDL, HDL, and triglycerides in the main? I defer to the authors and editors, but I suspect many readers would appreciate seeing the results for all four traits together.

Response: Thanks for the observations. Indeed, there was no reason for placing the TC in the S Figure and the other 3 traits in the main figure, beyond ensuring better visibility. Based on the reviewer's recommendation we have added all the four traits in Figure 5.

6. In the main text or in the legend of Figure 5, it would be helpful to report the GWAS sample size for each set of summary statistics.

Response: We agree that displaying sample sizes in the figure legend is useful. Therefore, based on the reviewer's recommendation we have added the sample sizes in the legend.

7. In the Discussion, the authors describe a look up of their SNPs in a very large GLGC GWAS. Are these unpublished results? If so, this fact should be specified. Otherwise, please provide a reference.

Response: Thank you for this observation. Since submitting the manuscript, the GLGC study has been published (Grahams et al. 2021) and we have added the reference in the current version of the manuscript.

8. Supplemental Table 8 appears to list incorrect p values for the METAL analysis (perhaps missing leading 0?)

Response: We thank the reviewer for pointing this out. There was a column shift, due to which the P-value column got replaced by sample size. We have amended this Supplementary Table (now S Table 10) accordingly.

REVIEWER COMMENTS (Round 2)

Reviewer #1 (Remarks to the Author):

Happy to authors.

Reviewer #2 (Remarks to the Author):

Thank you for addressing the comments.

Reviewer #3 (Remarks to the Author):

I appreciate the opportunity to review this revised submission of the manuscript, "Meta-analysis of ~25,000 continental Africans provides insights into the genetic architecture of lipid traits," by Choudhury and colleagues. In their revision, the authors have thoroughly addressed my prior comments, and I also found their comments in response to the reviewers to be very informative. Overall, this is an important paper with novel insights and sound methods. I recommend acceptance of the current revision.